# Progress in the Preparation and Applications of Microcapsules for Protective Coatings Against Corrosion

**DOI:** 10.3390/ijms26041473

**Published:** 2025-02-10

**Authors:** Shupei Liu, Jiajun Li, Yanchi Zhang, Xinfang Zhang, Yao Ding, Wenrui Zhang, Jinsong Rao, Yuxin Zhang

**Affiliations:** College of Materials Science and Engineering, Chongqing University, Chongqing 400044, China; 17862182689@163.com (S.L.); ljj804970863@163.com (J.L.); 15772046781@163.com (Y.Z.); 202109021146t@cqu.edu.cn (X.Z.); 13893404854@163.com (Y.D.); 19562219387@163.com (W.Z.); rjs@cqu.edu.cn (J.R.)

**Keywords:** organic polymer microcapsules, inorganic microcapsules, organic–inorganic hybrid microcapsules, anti-corrosion coating, research progress

## Abstract

The annual economic loss caused by corrosion accounts for about 2%~4% of GDP, which exceeds the sum of losses caused by fires, floods, droughts, typhoons, and other disasters. Coating is one of the most effective methods to delay metal corrosion. With the development of technology and the intersection of disciplines, functional microcapsules have been applied to anticorrosive coatings, but microcapsules are still being updated. To understand the application progress of microcapsules in anticorrosive coatings, the future development trend of microcapsules is analyzed. The preparation methods, physical and chemical properties, functional characteristics, and development trends of organic, inorganic, and organic–inorganic hybrid microcapsules are described, respectively, from the perspective of material and molecular characteristics. Simultaneously, the influence of microcapsules of different materials on the properties of organic coatings is proved by examples. In addition, the research status and future development trends of microcapsule composite coating are introduced in detail. Finally, the great advantages of organic–inorganic hybrid microcapsules modified by functional materials based on natural inorganic materials in improving the utilization efficiency of loaded active substances and prolonging the life of coatings are foreseen.

## 1. Introduction

The metal corrosion has caused great losses to the economies of various countries, even brought catastrophic accidents, wasted valuable resources and energy, and caused serious environmental pollution [1,2,3]. According to the Global Times, under the influence of corrosion, 10% of the steel produced in the world is consumed by corrosion, and the annual cost of anti-corrosion in the world is as high as 2.5 trillion US dollars, accounting for 3.4% of the global GDP [4,5,6]. Furthermore, the accidents caused by corrosion are even more shocking. For example, in 1981, the passenger plane b-737 in Taiwan Province, China suffered from serious intergranular corrosion in many parts of the fuselage, which led to an accident. The catalytic reforming unit of Ana Cortese Refinery of TESORO Company in the United States broke down due to high-temperature hydrogen corrosion on 2 April 2010, resulting in the serious consequences of seven employees’ deaths. On 2 March 2011, due to the aging equipment of the Fukushima nuclear power plant in Japan, embrittlement of the pressure vessel of the atomic furnace, and corrosion of the pressure suppression chamber, high-temperature nuclear fuel leaked, which caused incalculable losses to marine life and human beings. Therefore, many researchers have devoted themselves to developing new products with long-term corrosion protection and innovating new strategies to delay the corrosion of metal equipment.

In the past decades, researchers have conducted a lot of research on exploring ways to adjust or delay metal corrosion. At present, the commonly used metal anti-corrosion methods include metal surface treatment, corrosion inhibitors, coating protection, electrochemical protection, and so on [7,8,9]. Among them, the most commonly used method is to coat the surface of a metal substrate with macromolecular polymers to isolate corrosive media [10,11]. According to the different molecular characteristics of organic polymers, researchers designed a variety of polymer composite coatings, including epoxy coatings, vinyl coatings, and zinc-rich coatings, to extend the life of metal equipment. For example, Xu [12] and others prepared epoxy-based composite coatings with excellent thermal stability, hydrophobicity, and corrosion resistance by grafting hydrophobic molecular groups on the surface of SiO_2_ and modifying epoxy resin with hydrophobic SiO_2_. Wang [13] made full use of the excellent resistance of polyethylene (UHMWPE) molecules to develop an aluminum alloy protective coating. The corrosion thermodynamics and kinetics mechanism of composite coatings were studied by neutral salt spray and electrochemical analysis, which provided a new strategy for preparing high-density UHMWPE coatings. Similarly, Shen [14] and others designed a supramolecular heterojunction composite coating by grafting graphene oxide (GO) nanosheets onto highly conductive titanium dioxide (Ti_3_C_2_) and adding it to zinc-rich epoxy resin, which showed excellent barrier performance and cathodic corrosion inhibition.

Although with the continuous increase in polymer molecular weight, the compactness and corrosion resistance of the coating to metal substrate are significantly improved. However, with the demand for different functional coatings in many industries, it is no longer possible to improve the anti-corrosion performance of coatings by modifying resins with molecular technology or adding fillers, so the research and application of multifunctional coatings based on material protection has become a research hotspot [15,16,17]. To endow coatings with special functions, researchers integrate the advantages of physics, chemistry, and pharmacology, and combine microcapsule technology with coatings from the perspective of molecular design, thus endowing coatings with new functions [18,19,20]. For example, Meng et al. [21] prepared graphene/isophorone diisocyanate (IPDI) microcapsules with self-healing properties by in situ polymerization. Ruihan team [22] synthesized polyaniline (PANI)/epoxy resin copolymer with polyaniline (PANI)/epoxy resin copolymer as the core material by chemical oxidation method, and showed excellent self-repairing characteristics and synergistic anti-corrosion ability. In order to further improve the reaction characteristics of microcapsules, Xu et al. [23] were inspired by the self-healing mechanism of natural organisms, and successfully prepared dual self-healing multifunctional coatings by using multi-cavity microcapsules stimulated by temperature. With the urgent demand for anti-corrosion functional coatings in engineering applications, microcapsules provide a new strategy for coating functionalization and long-term anti-corrosion.

Microcapsules with different materials and preparation methods have been explored in many fields, such as anti-corrosion, medicine, catalysis, textile, active substance storage, and food preservation, and achieved remarkable results. The application of microcapsules in the field of anti-corrosion also shows new advantages. Commonly used antiseptic microcapsules are classified according to their material composition and molecular characteristics. As shown in Figure 1, according to the different development histories of microcapsules, the preparation, surface functionalization, corrosion resistance, and response characteristics of organic microcapsules (Figure 1A), inorganic microcapsules (Figure 1B), and organic–inorganic hybrid microcapsules with special response ability (Figure 1C) and their composite coatings are reviewed. Although with the improvement of technology, the durability and stability of some microcapsules can no longer meet the needs of high-performance materials in the current field, such as organic single-walled microcapsules, they are still of great significance in explaining the mechanism and development of microcapsules. In order to better sort out the characteristics of different microcapsules, the basic physical and chemical properties and characteristics of microcapsules are shown in Table 1, according to the different wall materials of microcapsules. It is of great significance to the preparation of new functional microcapsules and their application in the field of coatings [24,25,26,27].

## 2. Organic Microcapsules and Their Composite Coatings

Encapsulating active substances within organic polymer-based microcapsules is one of the most convenient and efficient methods for their protection and storage [48,49,50,51]. Therefore, adding microcapsules to the coating has great potential to enhance the physical properties of the coating and realize the synergy of chemical substances. As shown in Table 2 and Table 3, there are great differences in preparation methods of microcapsules with different functional characteristics, materials, and morphological structures, among which materials, microcapsule structure, and production cost are the best and most important factors in determining the preparation methods of microcapsules. Commonly used organic polymers for these microcapsules include polysulfone resin, polyurea (or polyaldehyde) resin, and epoxy resin [18,52,53]. As the characteristics of polymers are different, there are various methods to prepare organic polymer microcapsules, among which the most commonly used methods include interfacial polymerization, phase separation, extrusion, and spray drying. With the different requirements for microcapsules in coating anti-corrosion, biopharmaceuticals, food processing, and other fields and the progress of microcapsule preparation technology, researchers have developed double-walled microcapsules on the basis of traditional polymer single-walled microcapsules [54,55,56]. Therefore, according to the structural classification of organic microcapsules, polymer microcapsules can be mainly divided into single-walled and double-walled microcapsules (Table 2).

### 2.1. Preparation Method of Organic Single-Walled Microcapsules

#### 2.1.1. Complex Coacervation Method

As shown in Figure 2A, the complex coagulation method is a method of condensing two emulsions with different charges into capsules by electrostatic attraction [57,58]. According to the molecular characteristics and charge of wall materials, accurate adjustment of environmental pH value is the key to successfully preparing microcapsules by composite solidification method [59]. The microcapsules prepared by this method have the characteristics of high embedding efficiency, good biocompatibility, and slow controlled release. The preparation method has been widely used in the fields of drug delivery, food preservation, cosmetics, and other fields with strong demand for microcapsules [60]. As shown in Table 2, due to the different molecular structures of polymers, the properties of microcapsules usually vary. For example, in the research of Li et al. [61], tetrachloroethylene microcapsules were prepared with gelatin, sodium dodecyl sulfate (SDS), and sodium carboxymethyl cellulose (NaCMC) as raw materials by complexation and coagulation (Figure 2B). Microcapsules were gradually formed by the interaction between the protonated amino group of protein and the deprotonated carboxyl group of polysaccharide NaCMC, as a kind of water-soluble linear polymer produced by partially replacing the 2,3 and 6 hydroxyl groups of cellulose with hydrophobic carboxymethyl groups, making microcapsules more compact with better barrier and thermal stability. At the same time, in order to further prepare environmentally friendly microcapsule shells, Qi et al. [62] used the combination of free amino and carboxyl groups of Whey Protein Isolation (WPI) and high methoxyl pectin (HMP), and finally crosslinked with tannic acid (TA) to form a wall material encapsulating bioactive compounds (Figure 2C). However, compared with the wider interaction provided by WPI, the functionality of NaCMC becomes more limited.

Through decades of development, great progress has been made in the preparation of microcapsules by complex coacervation. With excellent dispersibility, high mechanical strength, and remarkable sustained-release effect, it is widely used in the fields of drug delivery, food processing, environmental remediation, preparation of functional materials, and so on. However, the complex preparation process and high cost limit its large-scale application in industry.

#### 2.1.2. Spray-Drying Process

Spray drying is a method to prepare microcapsules by rapidly drying atomized material fog in a high-temperature environment [63,64]. Spray drying mainly includes centrifugal, pressure, and airflow spray-drying methods [65]. This method can directly dry the emulsifier and solution into powder or granular products and can avoid complicated processes such as evaporation and pulverization. After spray drying, following emulsification, the microcapsules have great differences in functions due to the different molecular structures of emulsifiers. Therefore, this method is widely used in biopharmaceutical, chemical, health-care, food, and other industries because of its simple process and easy mass production.

For example, Zhang [66] et al. used gelatin/microcrystalline cellulose (GLT/MCC) complex as a co-emulsifier, and maltodextrin as wall material, emulsified and spray-dried to prepare Zanthoxylum bungeanum essential oil microcapsules. GLT forms a colloidal structure by hydrogen bonding and electrostatic interaction between abundant polar amino acid residues and water molecules, and MCC provides better physical stability for the emulsion. Tao et al. [67]. prepared hydrophobic microcapsules of grape seed oil loaded with pterostilbene (PTE) by spray drying. The methoxy group and stable phenolic structure of pterostilbene (PTE) enable it to enhance its bioavailability, which makes the aroma of grape seed oil retained by a large degree. Rodríguez-Cortina et al. [68] prepared camellia seed oil (SIO) microcapsules by spray drying and compared them with those obtained by freeze-drying technology. It is found that spray-drying microcapsules have higher embedding efficiency and are more economical and effective than freeze-drying microcapsules.

The advantages of spray drying are evident. First of all, the drying conditions are highly adjustable, the process is rapid, and the product quality can be effectively controlled. Second, spray drying is a straightforward and operationally simple method, offering high production efficiency and scalability for mass production. Moreover, the equipment allows for flexible adjustments to meet specific requirements, including control over humidity, particle size, solubility, and the activity of encapsulated substances. However, the spray-drying equipment is usually large, which requires a lot of initial investment and will produce a lot of heat loss during the drying process. Generally speaking, spray drying may not be the most energy-saving method, but the authors believe that the preparation of microcapsule carriers on a large scale has unparalleled advantages, especially suitable for processing heat-sensitive materials that need to be dried quickly. With the continuous progress of technology and the expansion of application fields, spray drying is expected to play an increasingly important role in various industries.

#### 2.1.3. In Situ Polymerization

In situ polymerization is a method in which reactive monomer or its soluble prepolymer and catalyst are all added into the dispersed phase (or continuous phase) [69]. Because monomer is soluble in a single phase and its polymer is insoluble in the whole system, polymerization takes place on the core material of the dispersed phase [70]. In situ polymerization can directly form microcapsules in the reaction process, which reduces the subsequent processing steps and improves production efficiency. In addition, microcapsules with uniform particle size and uniform distribution can be obtained by controlling polymerization conditions to ensure their consistency in application. In situ polymerization usually uses water phase or other environmentally friendly solvents, which reduces the use of organic solvents and conforms to the principle of green chemistry. In addition, in situ polymerization has strong adaptability and has been widely used in medicine, food, cosmetics, and other fields [71].

For example, Xue [72] et al. encapsulated sodium silicate restorer in polyurethane (PU) microcapsules by in situ polymerization, and prepared single-wall self-repairing polyurethane microcapsules. Zhang [73] and others prepared a network structure formed by condensation reaction between urea and formaldehyde through in situ polymerization, which has strong hydrogen bonding and crosslinking density. However, its formation process requires high reaction temperature and acidic conditions. In order to further reduce the cost and improve the loading rate of active substances, Christos [74]. prepared epoxy resin-loaded microcapsules with polyurea formaldehyde (PUF) shell by in situ polymerization. Polyurea and formaldehyde form a flexible molecular crosslinking network through polymerization. The amino group in the polyurea group reacts with isocyanate to form a loose structure, which makes microcapsules more controllable. The biocompatibility and controlled release characteristics of microcapsules formed by polyurea formaldehyde significantly improve the embedding efficiency and thermal stability, and the self-repairing ability of composite coatings is significantly improved.

Nevertheless, there are still some problems in the strategy of preparing microcapsules by in situ polymerization. For example, temperature control is difficult. Secondly, in the process of polymerization, side reactions may occur, affecting the purity of the final product. It is difficult to scale up, and heat-sensitive substances may be degraded due to high temperatures or long-term reactions, which will affect their activity and effect.

#### 2.1.4. Single Coagulation Method

The condensation method is a method to reduce the solubility of polymer by adding coagulant into the solution of polymer capsules and solidifying it into capsules [75]. The key to this method is to use strong hydrophilic electrolytes or non-electrolytes as coagulants to destroy the solvation of gelatin molecules, thus reducing their solubility and making the polymer precipitate and shrink into capsules. At the same time, microcapsules with specific functions and applications can be prepared through appropriate process control. For example, Lazko et al. [76] encapsulated the dispersed oil phase (hexadecane) by a single condensation method. It is found that the long-term stirring of the reaction medium after solidification can significantly increase the unevenness of the particle size distribution of microcapsules. Weiβ et al. [77] prepared hydroxypropyl methylcellulose phthalate (HPMCP) polymer microcapsules in 20% (*w*/*w*) sodium sulfate solution by simple coagulation method. As a water-soluble cellulose derivative, HPMCP is stable in an acidic environment because of its hydroxypropyl and methyl groups in its molecular structure. HPMCP microcapsules are suitable for drug and health-care product delivery. However, it is easily hydrolyzed under high humidity, which affects its long-term stability and controlled release performance. Li et al. [78] prepared microcapsules loaded with citral essential oil (CEO) by single coagulation method. As a natural essential oil, citral essential oil (CEO) has strong antibacterial and antioxidant properties. The experimental structure shows that microcapsules have good microstructure and the ability to reduce and control CEO release.

Although the technical principle of this method is simple, it has special requirements for the molecular structure and stability of materials. At the same time, compared with complicated coagulation methods, it has some obvious shortcomings. For example, microcapsules prepared by a single coagulation method have obvious shortcomings such as relatively low embedding efficiency and short sustained-release life. Therefore, although the single molecule coacervation method is a feasible method to prepare microcapsules, the disadvantages of the complex preparation process and slightly low drug loading still need to be overcome.

#### 2.1.5. Interfacial Polymerization

Interfacial polymerization refers to the method in which two monomers with high reactivity are dissolved in two incompatible solvents, and irreversible polycondensation occurs at the interface of two liquid phases to form microcapsules [79,80]. Polymers produced by polymerization have a high yield, many kinds, and a wide application range. Among the interfacial polymerization methods, there are interfacial polymerization reactions and interfacial addition reactions, among which interfacial condensation reaction is the most widely used, and there are few studies about synthesizing some materials through interfacial addition. There are two basic types of interfacial condensation processes. One is that different small molecular substances (such as isocyanate, polyphenol, and polyol) are dissolved in different liquid phases at room temperature, and react at the interface to form microcapsules by centrifugal force. Secondly, in the interfacial condensation reaction industry, all small molecules dissolved in the oil phase can be polymerized at the interface by raising the temperature. When the small molecular substance is aminoplast, the interfacial polymerization is promoted by heating or adding active acidic substances. Therefore, interfacial polymerization technology plays an important role in the preparation of fiber raw materials, microcapsule materials, membrane materials, and new materials [81].

For example, Liang et al. [82] prepared nano-silicon carbide (SiC)-doped long-chain fatty acid ester (tetradecyl octadecanoate (TO)) phase-change microcapsules by interfacial polymerization (Figure 3A). The release control accuracy of microcapsules was significantly improved. Liu et al. [83] prepared a new formaldehyde-free microcapsule with polyurethane-urea (PUU) as the shell by interfacial polymerization. Polyurethane-urea has good mechanical strength, thermal stability, and chemical stability (Figure 3B). The amino group and urea group in its molecule can form a strong cross-linked structure through hydrogen bonds and covalent bonds, which endows microcapsules with high physical strength as well as durability and is suitable for long-term controlled release systems. In order to further improve the response of microcapsules to external stimuli. Zhao et al. [84] prepared microcapsules with polyurethane (PU) as shell by interfacial polymerization, which can trigger the release of 2-dibutylamine-4,6-dimethylmercapto-1,3,5-triazine (DB) at a predetermined temperature (Figure 3C). Microcapsules with controllable glass transition temperature were successfully prepared by using mixed solvents as an organic phase to control the diffusion of common monomers during interfacial polymerization.

Although the method of preparing microcapsules by interfacial polymerization has its unique advantages, such as preparing microcapsules with smooth surface and small particle size, it also has some shortcomings. Compared to traditional methods, interfacial polymerization involves a relatively complex process, potentially increasing production costs and operational challenges. Additionally, while this method enables the fabrication of microcapsules with smooth surfaces and small particle sizes, its low synthesis efficiency may limit its suitability for large-scale production.

**Table 2 ijms-26-01473-t002:** This table reports the advantages and potential values of different organic single-walled microcapsules and their anticorrosive composite coatings.

Type Single-Walled Microcapsule	Preparation Method	Advantages of Method	Disadvantages of Method	Advantages of Composite Coating	Disadvantages of Composite Coating
Single-walled microcapsule	Single coagulation method [76,77,78,85,86]	● Excellent uniformity and dispersibility; ● High mechanical strength; ● Better sustained-release effect.	● Preparation process is relatively complicated; ● Drug loading is slightly lower; ● Encapsulation efficiency of microcapsules is low.	● Excellent self-repair and corrosion inhibition performance; ● The filler has good dispersibility and high coating density; ● The active substance acts quickly.	● The active substance has a fast slow-release speed and short service life; ● The microcapsules in the coating are easy to break and fail, and the stability is poor; ● Low utilization rate of active substances.
	Complex coacervation method [59,60,61,62]	● Loading efficiency of active substances is high; ● Stability is obviously enhanced; ● Strong adaptability; ● Process is relatively simple.	● Preparation process is complicated; ● High cost.		
	Spray-drying process [66,67,68]	● High yield; ● High efficiency and simple control.	● Equipment used in spray drying is bulky; ● High investment; ● Heat loss in the drying process is large.		
	In situ polymerization [71,72,73,74,87]	High production efficiency and uniform particle size.	● Temperature control is very difficult; ● During the polymerization process, side reactions may occur.		
	Interfacial polymerization [82,83,84,88]	● Reaction conditions are mild; ● Mechanical strength is high.	● Difficult to control the reaction rate; ● Purity of raw materials and equipment is high.		

### 2.2. Organic Single-Walled Microcapsules in Anti-Corrosion Coating

The early research on polymer microcapsules mainly focused on the preparation of single-walled microcapsules, encapsulation, and storage of active substances with poor environmental tolerance [18,51]. With the continuous progress of microcapsule technology and the increasing demand for marine corrosion protection for coating functions, single-walled microcapsules with functional active substances are gradually widely used in the coating field and endowed with multifunctional coatings. Up to now, polymer single-walled microcapsules have been widely used in the development and preparation of marine anti-corrosion, antifouling, self-repair, and self-cleaning functional coatings [89].

To improve the anti-corrosion performance of polyurea resin coating and endow the coating with anti-fouling and anti-scaling ability, Liu et al. [90] used α-cyanoacrylate and hydroxyethyl diphosphonic acid (HEDP) to prepare porous single-walled microcapsules loaded with HEDP by simple spray-drying method. As shown in Figure 4A, porous microcapsules were added to the polyurea coating, and the excellent corrosion resistance of the composite coating was proved by electrochemical and salt spray tests. In order to further improve the corrosion resistance of microcapsules, Zhao [91] and others prepared polysulfone (PSF) microcapsules loaded with propane-1,2,3-triol corrosion inhibitor by interfacial polymerization and discussed the corrosion resistance and protective performance of the self-repairing coating of PSF microcapsules. Studies show that microcapsules have excellent healing ability and corrosion inhibition function for microcracks (Figure 4B). In order to further improve the utilization efficiency of corrosion inhibitors, Liu et al. [92] used polysulfone, stearic acid, and corrosion inhibitor to prepare antiseptic microcapsule filler with pH response by in situ polymerization. Due to the existence of stearic acid, the filler can remain relatively stable under neutral and acidic conditions. When oxygen inhalation corrosion occurs, the pH value of the surrounding environment increases, and the stearic acid on the surface of microcapsules decomposes rapidly, releasing corrosion inhibitor (Figure 4C) [93,94]. In addition, the team added pH-responsive microcapsules to polyurea coating to verify the corrosion inhibition ability of microcapsules. The experimental results show that the impedance modulus of the composite coating with pH-responsive microcapsules is about two orders of magnitude higher than that of the blank polyurea coating after testing in 3.5 wt.% NaCl solution for 30 days.

In addition, the methods of preparing antiseptic microcapsules by single coagulation and complex coacervation have also been widely studied. For example, Dong et al. [95] used modified silica nanoparticles as stabilizers and 2-mercaptobenzothiazole as corrosion inhibitor, and prepared intelligent polyaniline microcapsules with pH response and corrosion inhibition characteristics by single coagulation method (Figure 4D). Due to the combination of the characteristics of metal passivation of polyaniline (PANI) microcapsules and the release of MBT on demand, the synergistic anti-corrosion of active solid filler (PANI) and slow-release corrosion inhibitor (MBT) was realized, thus providing double anti-corrosion protection for low carbon steel. Wang et al. [96] used oxygen plasma to embed carbon nanotubes (OPCNTs) to improve the micro-mechanical behavior of the microcapsule shell, and prepared a new type of urea-formaldehyde isocyanate microcapsules by complex coacervation. Yin et al. [97] synthesized epoxy-based microcapsule materials with self-healing ability by further optimizing the coagulation conditions and using latent curing agents containing epoxy resin and urea-formaldehyde micro-components.

The above research has proved that the single microcapsule can improve the barrier, corrosion resistance, and self-repair ability of the coating and its application prospect. However, due to the limitations of organic polymers and the single protection of single-walled microcapsules to active substances or inhibitors, organic single-walled microcapsules show poor stability. Self-repair and early damage or rupture of slow-release microcapsules are some of the unstable factors of organic single-walled microcapsules.

### 2.3. Organic Double-Walled Microcapsules and Preparation Method

The preparation technology of double-walled microcapsules has become a hot topic because of their excellent compactness, mechanical properties, and higher stability [36,98]. On the basis of single-walled microcapsules, it is the most commonly used method to form the second capsule wall by secondary wall-building to form polymer polycondensation deposition. At present, the commonly used preparation methods of organic double-walled and multi-walled microcapsules include solvent evaporation combined with sol–gel method, in situ polymerization combined with surface loading method, interfacial polymerization combined with in situ polymerization, and the use of composite emulsifiers (Table 3).

#### 2.3.1. Multiple Sol–Gel Method

The sol–gel method is a widely used technique for preparing microcapsules and is regarded as an advanced approach for constructing double-layer wall structures via a two-step sol–gel process [99]. The first step of this method is to select suitable inner wall materials and prepare single-walled microcapsules containing core materials. The second layer of wall material was deposited on the surface of single-walled microcapsules by sol–gel method and double-walled microcapsules were successfully prepared. At the same time, the heat treatment process can further promote the solidification of the gel network and improve the thermal stability of microcapsules. The double-walled microcapsules prepared by this method have good thermal stability, high encapsulation efficiency, and excellent mechanical properties. In addition, the controllable size, morphology, and chemical properties are the advantages of this method in many fields such as self-lubrication, phase-change energy storage, temperature-sensitive discoloration, and so on [100].

For example, Charlie et al. [101] synthesized deep eutectic solvent (DES) microcapsules using a non-aqueous sol–gel method via oil-in-oil emulsions and silane monomer polycondensation (Figure 5A). The microcapsules were prepared through a cross-linking reaction between tetraethoxyorthosilicate and polydimethoxysiloxane, catalyzed by formic acid. The resulting silica shells exhibited excellent chemical stability and physical durability, making them easier to handle compared to the viscous bulk DES. This method demonstrates the advantages of combining oil-in-oil emulsions with the sol–gel approach for the production of stable and reusable DES-based microcapsules. Zhou [102] made full use of the strong self-assembly characteristics of dopamine hydrochloride, and prepared dopamine hydrochloride-SiO_2_ double-wall self-repairing microcapsules by secondary wall construction method (Figure 5B). Qian et al. [103] prepared double-walled microcapsules with n-disaccharide as the core and Fe_3_O_4_ and SiO_2_ as the inner shell by sol–gel method (Figure 5C), improving the stability and temperature control ability of the phase-change material.

Although the preparation of double-walled microcapsules by the sol–gel method shows unique advantages. However, in addition to the shortcomings of the complex technology of double-walled microcapsules. The disadvantages of raw materials (such as metal alkoxides), such as high cost, environmental pollution, long drying cycle, and possible collapse and contraction of the shell during drying, should also be paid attention to. Nevertheless, the sol–gel method is still of great value in preparing double-walled microcapsules with specific functions, especially in improving the thermal stability and mechanical strength of microcapsules.

#### 2.3.2. Combination of In Situ Polymerization Method and Surface Loading Method

To precisely control the inner and outer shells, in situ polymerization was combined with the surface loading method to realize the controllable preparation of double-walled microcapsules [104]. In situ polymerization allows precise control of the chemical composition and physical structure of inner wall materials to meet specific application requirements. The physical barrier properties of microcapsules can be further enhanced by the surface loading method, and the stability and controllability of their release characteristics in complex environments can be improved. The double-walled microcapsules prepared by combining these two methods not only have excellent physical and chemical stability, but also can adjust the composition and thickness of double-walled materials according to needs, so as to realize accurate regulation of the release behavior of core materials. Through systematic experimental design and optimization, the effects of different wall material combinations on the properties of microcapsules were further discussed, which provided the scientific basis for the design and application of microcapsules.

For example, Feng et al. [105] prepared poly-dopamine/polyaniline (PDA/PANI) microcapsules by hydrofluoric acid etching method, and then filled benzotriazole poly-dopamine (BTA) inhibitor into the microcapsules by in situ polymerization method, thus preparing BTA@PDA/PANI microcapsules. The phenolic hydroxyl group of benzotriazole and the amino group of water can form a powerful composite network through hydrogen bonding, π-π interaction, and electrostatic interaction, which enhances the stability and activity of microcapsules. However, PDA/PANI is hydrophilic, which may lead to early release or membrane rupture in some humid environments, limiting its effectiveness in some long-term stable applications. In order to improve the mechanical stability and environmental adaptability of microcapsules, Li et al. [106] first prepared polysulfone (PSF) microcapsules with high-temperature resistant lubricating oil BIO_3_O as the core by solvent evaporation, and then activated and modified its surface by in situ self-assembly of poly-dopamine and graphene oxide. The surface modification of graphene prevented the softening and decomposition of PSF microcapsules. The aromatic ring and sulfhydryl structure in pounds per square foot (abbreviation of pounds per square foot) give microcapsules strong corrosion resistance and high-temperature adaptability. By coating BIO_3_O, a high-temperature resistant lubricating oil, microcapsules have good thermal stability and excellent tribological properties.

Although in situ polymerization and surface modification have shown great potential in the field of microcapsule preparation, their limitations cannot be ignored. These shortcomings stem from the complexity of the preparation process itself and the diversified needs of application scenarios and also reflect the high sensitivity of technical aspects such as temperature, pH value, and stirring speed. This dependence on process accuracy makes large-scale production face great challenges. At the same time, the material distribution is uneven and insufficient in the process of surface modification, which shows low activity. What is more prominent is that the chemical reagents involved in the process of modifying the microcapsules that are not completely encapsulated may have an impact on the core substances, and may even damage the integrity of the core functional substances. For solving these problems, process optimization and material innovation still need to be further studied, so as to reduce the impact on the environment while giving consideration to efficiency and function, so as to promote the development of microcapsule technology in a more mature and diversified direction.

#### 2.3.3. Combination of Interfacial Polymerization and In Situ Polymerization

The preparation of double-walled microcapsules by interfacial polymerization and in situ polymerization is an innovative technology combining two chemical methods [48]. The double-walled microcapsules prepared by combining these two methods not only realize the uniform composite of inner and outer wall materials but also realize the precise regulation of the size, morphology, and properties of microcapsules by accurately controlling the polymerization conditions. This method effectively improves the physical and chemical stability of microcapsules and enhances their application potential in complex environments [107]. In addition, the process has good expansibility and is suitable for a variety of materials and mass production, which provides new possibilities for the application of microcapsule technology in the fields of drug delivery, food additives, coating functionalization, and so on.

Many researchers have explored the effects of different wall materials on the properties of microcapsules by studying molecular characteristics, system design, and optimization, which provided the scientific basis for the design and application of microcapsules. For example, Yuan et al. [108] made use of the self-repairing advantage of the rapid polymerization of isophorone diisocyanate to prepare phenolic aldehyde (PF)/polyurethane (PU) double-walled microcapsules by interfacial polymerization and in situ polymerization (Figure 6A). The double-wall composite structure formed by the combination of phenolic aldehyde and polyurethane improves the stability and self-healing efficiency of IPDI due to the synergistic effect of its cross-linking structure. Sun [109] successfully synthesized liquid 4,40-bis-methylcyclohexane diisocyanate double-shell microcapsules with excellent water resistance in oil-in-water emulsion by combining interfacial polymerization and in situ polymerization (Figure 6B). In order to improve the response of microcapsules to external stimuli, Li et al. [110] transferred ionic liquid ([BMIm]PF_6_) into multi-wall microcapsules by county-level joint method, which realized the self-lubrication of microcapsules, and the prepared composite coating showed ultra-low friction coefficient and wear rate (Figure 6C).

The above research has proved the scientific research value and application prospect of preparing double-walled microcapsules by combining interfacial polymerization with in situ polymerization. However, there is still room for improvement in this emerging technology. First of all, interfacial polymerization occurs at the interface of different phases, and there is a possibility that side reactions will destroy the activity of the core material. Secondly, in situ polymerization, the organic matter often volatilizes incompletely, which leads to product pollution and affects capsule characteristics. Moreover, the flocculation problem of microcapsule products in dispersion media cannot be ignored. Therefore, although the double-walled microcapsules prepared by the combination of interfacial polymerization and in situ polymerization have many potential advantages, these shortcomings need to be overcome in practical applications in order to realize their wide application in various fields.

#### 2.3.4. Composite Emulsifier

Double emulsification is an effective technology to prepare double-walled microcapsules and encapsulate active ingredients by twice emulsification [111]. Different molecular structures in composite emulsifiers make microcapsules have different properties. The research shows that the composite emulsification technology is realized by changing the force between oil droplets and the competitive adsorption of the composite emulsifier. Many researchers have studied this and achieved fruitful results. For example, Ma [112] and others used sodium dodecyl benzene sulfonate with hydrophilic and hydrophobic groups and polyvinyl alcohol (PVA) as composite emulsifiers to prepare microcapsules with anti-corrosion and anti-fouling functions. In order to further control the particle size of microcapsules, Monika Devi [113] and others prepared stable microencapsulated anthocyanins by composite emulsification. The minimum particle size distribution of microcapsules is 24.07~71.33 μm m. In addition, Koo et al. [114] emulsified a multiphase mixture containing liquid prepolymer and magnetic nanoparticles (MPs) in chloroform solution to form a liquid consisting of chloroform core and MPs/polymer shell. At the same time, it is proved that the active substance loading rate of this method is as high as 75 wt%. Improving the stability of emulsion and the loading rate of active substances is the most obvious advantage of composite emulsification. However, the relatively complex process and harsh emulsification and reaction conditions of the compound emulsification method not only increase the cost but also may pose a threat to safe production. In addition, composite emulsifiers can improve the stability of the emulsion, but they may affect other characteristics of microcapsules, such as release characteristics and biocompatibility. Moreover, the preparation and application of composite emulsifiers may require special technology and equipment, which limits its application in large-scale industrial production. Therefore, although the composite emulsifier has certain advantages in preparing double-walled microcapsules, it is necessary to overcome these shortcomings in practical application in order to realize its wide application in various fields.

**Table 3 ijms-26-01473-t003:** This table reports the characteristics of different organic double-walled microcapsules and their effects on anti-corrosion composite coatings.

Type Single-Walled Microcapsule	Preparation Method	Advantages of Method	Disadvantages of Method	Advantages of Composite Coating	Disadvantages of Composite Coating
Organic double-walled microcapsules	Sol–gel method [36,102,103]	● It can be prepared at room temperature; ● The product has high purity and good uniformity; ● The product shape and size are easy to control; ● Environment-friendly.	● Higher cost; ● Organic matter in raw materials may also be harmful to human health; ● Long coagulation time; ● Complex process and poor compactness.	● The stability is significantly increased; ● The coating life is significantly enhanced; ● The slow-release effect is good, and the utilization efficiency of corrosion inhibitors or active substances is significantly increased.	● The comprehensive cost of coating increases; ● The coating process is complicated; ● The risk of environmental pollution increases with the increase in organic solvent usage.
	In situ polymerization-surface loading method [105,106,115]	● Enhanced stability; ● Improve drug loading; ● Controlled release ● Reduce monomer loss; ● Good dispersibility.	● The preparation process is complicated; ● Higher cost; ● Limitation of reaction conditions; ● Poor compatibility with resin; ● Pollute the environment.		
	Interfacial polymerization-in situ polymerization combination [108,109,110]	● A high degree of structural controllability; ● Efficient material utilization rate.	● Poor dispersion; ● Temperature and pH are difficult to control; ● Low production efficiency.		
	Composite emulsification method [112,113,114]	● The preparation process is relatively simple; ● The active substance has a large load, mild environment, and strong adaptability.	● Stability of difference; ● Low production efficiency; ● High viscosity and difficult preparation.		

### 2.4. Application of Organic Double-Walled Microcapsules in Anticorrosive Coatings

As we all know, adding self-repairing microcapsules to the coating is an effective method to solve the cracks and damage to the coating. Although the traditional single-walled microcapsules can self-repair microcracks to a certain extent, they still have the disadvantages of poor system stability, slow curing film formation speed, and high catalyst cost. To improve the stability of active substances stored in microcapsules, so as to improve the self-repair ability of microcapsules, some researchers have carried out research on double-walled microcapsules [36]. Compared with single-wall microcapsule, the new double-wall microcapsule can make a curing agent and repairing agent that exists in the inner and outer layers of the microcapsule core at the same time, which improves the mechanical properties of the shell, solves the problem of rapid crack repair, and provides a new scheme for the stable self-repair of the coating [116].

To further improve the stability of microcapsules, Ye et al. [117] prepared bifunctional self-healing microcapsules (PSG/dual mcs/ER) by a two-step casting method (Figure 7A). The experimental results show that epoxy resin microcapsules coated with epoxy resin-n-butyl alcohol ether have a uniform spherical shape, and the embedding efficiency is 90.42%. At the same time, microcapsules have good thermal stability, acid resistance, alkali resistance, and salt resistance. Zhang [118] et al. successfully synthesized microcapsules containing 4,4′-bis-methylcyclohexane diisocyanate (HMDI) by the microcapsule method (ES-IP), combining electrospinning and interfacial polymerization (Figure 7B). The test shows that HMDI microcapsule (HMDI-MCs) has good dispersibility and adjustability in the coating, and its core material content is as high as 93%. In order to further improve the loading capacity of corrosion inhibitor microcapsules, Feng et al. [105] prepared benzotriazole (BTA)/dopamine/polyaniline (BTA@PDA/PANI) microcapsules as shown in Figure 7C by hard template method, in situ polymerization, and vacuum impregnation. The BTA loading capacity of microcapsules exceeds 20 wt%.

The stability of stored active substances is significantly improved due to the protection of polymer double-walled microcapsules. Therefore, the preparation of double-walled microcapsules provides a strong guarantee for the storage of active substances and the stable release of functional substances. It is worth noting that organic polymers are still the main raw materials for preparing double-walled microcapsules, and most double-walled microcapsules still have the inherent defects of organic materials (poor thermal stability, intolerance to organic solvents, easy aging, etc.) [119,120,121]. Therefore, the preparation of double-walled microcapsules with physical and chemical stability and strong tolerance needs further study.

## 3. Inorganic Microcapsules and Anti-Corrosion Coatings

Organic microcapsules and inorganic microcapsules have their own unique advantages and disadvantages in the research and application of microcapsule technology [119,122]. However, with the increasing demand for higher-performance materials, inorganic microcapsules have attracted more and more attention because of their unique advantages. Although organic microcapsules are excellent in drug loading capacity and release control, their poor tolerance has limited their application in special conditions. As shown in Table 4 and Table 5, inorganic microcapsules with excellent acid and alkali resistance, chemical resistance, and thermal stability have been sought after by many researchers [40,123]. Usually, common inorganic microcapsules are mesoporous silica, halloysite, hollow carbon spheres, diatomite, and so on [124,125]. Due to its excellent stability, biocompatibility, and controlled release performance, inorganic microcapsules have gradually become the focus of microcapsule research. Through further exploration and research, people have explored some new inorganic microcapsules, such as layered double hydroxide (LDH) microcapsules, which have excellent ion exchange performance and chemical stability, making them good application prospects in drug delivery, environmental remediation, and other fields. Although the economy of emerging inorganic microcapsules still needs to be studied, its unique characteristics can still make it widely used [126,127].

### 3.1. Classification of Inorganic Microcapsules

#### 3.1.1. Traditional Inorganic Microcapsules

As shown in Table 4, traditional inorganic microcapsules have shown wide application potential in the fields of drug delivery, catalysis, sensors, and environmental remediation. Usually, traditional inorganic microcapsules include mesoporous silica, halloysite, diatomite and hollow carbon nanotubes.

Mesoporous silica (such as MCM-41 and SBA-15) has been widely concerned because of its highly ordered mesoporous structure and excellent specific surface area (usually greater than 1000 m/g) and porosity. Mesoporous silica has excellent chemical stability and can resist the corrosion of many acids, bases, and organic solvents, so it still maintains efficient functionality under extreme conditions. In addition, the surface can be chemically modified to enhance the interaction with drugs or functional molecules, so as to improve the drug-carrying capacity and the accuracy of release control. For example, as shown in Figure 8A, Yu et al. [128] encapsulated biomacromolecule active enzyme in composite microcapsules of polyelectrolyte and SiO_2_, which significantly improved the immobilization rate and enzyme loading. Bchellaoui [129] and others formed highly monodisperse mesoporous silica microcapsules with a diameter of 10 microns by better controlling the diffusion of silica precursor sol in the surrounding perfluorooil phase during the formation of silica, thus controlling the particle size within the micron range (Figure 8B). As shown in Figure 8C, Yang et al. [130] proposed a new method to prepare mesoporous silica microcapsules by loading silver nanoparticles (AgNPs) on the inner wall of the mesoporous silica shell. The unique structure of SiO_2_ microcapsules can effectively avoid the aggregation of traditional silver colloid systems and slowly release silver ions to induce antibacterial activity. AgNPs @ silica microcapsules have the advantages of high specific surface area, high antibacterial activity, and easy recovery, and are an ideal green and efficient antibacterial agent.

Halloysite is a layered silicate mineral. Its unique layered structure endows it with a large specific surface area and unique ion exchange ability, which makes it widely used in the fields of metal anti-corrosion, chemical catalysis, and environmental restoration. The natural source and good biocompatibility of halloysite make it an ideal choice for environmental protection materials, suitable for drug delivery systems and health-care products carriers. In the field of corrosion protection and catalysis, halloysite also shows significant application potential. In addition, its layered structure can be modified by physical or chemical methods to enhance its functionality and adaptability, thus improving its performance in a variety of application scenarios. For example, Wang et al. [131] used hollow HNTs as a nano-container loaded with ionic liquid, and reduced the thermo-mechanical deformation of the composite through HNTs/Au hybridization synergy (Figure 8D). Finally, the ionic liquid 1-hexyl-3-methylimidazolium bis (trifluoromethylsulfonyl) imine ([HMIm][NTf_2_]) was encapsulated to obtain an adaptive friction film and reduce the wear effect. Fahimizadeh et al. [132] loaded yeast extract (YE), a common microbial nutrient, into kaolin clay nanotubes (HNTs) and wrapped it with pseudobacillus spores in calcium alginate microcapsules (Figure 8E). The encapsulation of HNTs loaded with YE improves the resistance of microcapsules to the cement environment, promotes the formation of minerals, realizes the best self-healing of concrete, and prolongs the service life of concrete structures. Liu et al. [133] prepared organic phase-change material (PCM) microcapsules containing kaolin nanotubes (HNTs) by emulsion copolymerization (Figure 8F). The addition of kaolin improved the thermal storage efficiency and temperature adjustment ability of PCM microcapsules and also made CPM microcapsules have a good flame-retardant effect.

Diatomite is a kind of natural diatom that remains, with the advantages of high specific surface area, large pore volume, and low density. Because of its remarkable adsorption characteristics and low density, it stands out in filtration separation technology. Diatomite is rich in pore structure, which can effectively capture and release a variety of molecules, and is widely used in agriculture, the food industry, and drug delivery. For example, as shown in Figure 9A, Sun et al. [134] prepared melamine-urea-formaldehyde (MUF)-loaded phase-change diatomite microcapsules (microPCMs) by in situ polymerization. The addition of diatomite significantly improved the heat storage capacity of MUF and the overall performance of microcapsules. The authors of [135] successfully synthesized fluorosilane-modified hollow mesoporous diatom bio-silicone oil-loaded microcapsules (Oil@HMSNs/DE-F) with diatomite as hard template and dimethyl siloxane oil and hollow mesoporous silica as core materials (Figure 9B). The formed microcapsules have high oil-loaded capacity and oil-loaded stability, which makes the composite coating have excellent mechanical properties and tribological properties. Peng et al. [136] used poly-dopamine (PDA) and carboxymethyl chitosan (CMCS) to construct a nasal delivery carrier to modify diatomite bio-silica (DB) comprehensively (Figure 9C). The obtained microcapsules of DBB@PDA-CMCS (DBBPC) have high antibacterial activity and biological safety and can inhibit the complications of bacterial rhinitis.

Hollow carbon nano is a kind of carbon microcapsule, which not only has excellent loading capacity of microcapsules, but also shows excellent mechanical properties. It has been used to package and transport various functional molecules and has shown good drug-carrying ability and biocompatibility. For example, Xu et al. [137] used calcium carbonate (CaCO_3_) particles as a solid template to prepare hollow microcapsules of carbon nanotubes by electrostatic self-assembly (Figure 9D). The hollow microcapsules basically remained spherical, and the carbon nanotubes were uniformly fixed in the polyelectrolyte layer. Bae et al. [138] prepared carbon microcapsules containing Si-CNT nanocomposites (Si-CNT@C) by two-step polymerization (Figure 9E). The obtained Si-CNT@C microcapsules have internal free space, which can accommodate the volume expansion/contraction of silicon nanoparticles during lithium-ion charging and discharging. In addition, carbon nanotubes can provide higher electrical connections for crushed silicon nanoparticles, thus improving the performance of lithium-ion cathode. Xu [139] and others explored a new method of assembling single-walled carbon nanotubes (SWCNT) microcapsules by spray-drying and solvent-exchange technology. Compared with the traditional layer-by-layer assembly method, the preparation of carbon nanotube microcapsules by using spray-dried water-soluble templates can simplify the sacrifice process (Figure 9F). Inorganic microcapsules have attracted a lot of researchers’ attention because of their high chemical and thermal stability, controllable particle size, and excellent ability to protect active substances. However, organic matter is still an important substance to realize the functional activity of microcapsules. Therefore, it is difficult for a single inorganic microcapsule shell to realize intelligent release and effective utilization of load. It is an important trend to make full use of the excellent tolerance of inorganic capsule-free skeletons, integrate organic functional active substances, and give microcapsules special functions.

**Table 4 ijms-26-01473-t004:** The following table shows the characteristics of traditional inorganic microcapsules (mesoporous SiO_2_, diatomite, halloysite, etc.) and their effects on the anti-corrosion performance of coatings.

Type	Microcapsule Species	Preparation Method	Advantages of Microcapsules	Disadvantages of Microcapsules	Advantages of Composite Coating	Disadvantages of Composite Coating
Traditional inorganic microcapsules	Diatomaceous earth [134,135,136]	Natural acquisition	● High specific surface area; ● Natural reserves are large and easy to obtain; ● Low cost.	● Functionalization difficulty; ● The shape and size are uncontrollable; ● Uneven particle size; ● Insufficient completeness and purity.	● Enhance the barrier property of the coat; ● Reduce coating cost; ● Lighten coating weight; ● Give the coating other functions, such as fire prevention, sound insulation, and so on.	● Excessive addition of diatomite may lead to a decrease in coating adhesion; ● Lack of toughness.
	Attapulgite [128,129,130]	Natural acquisition or artificial synthesis	● High specific surface area; ● Stability analysis of structures; ● Environmental pollution-free.	● Low active material loading; ● The manual preparation process is complicated.	● Enhance the mechanical properties of the coating; ● Improve the barrier ability of the coating; ● Excellent sustained-release performance; ● The adhesion of the coating is significantly enhanced; ● Anti-impact and wear, suitable for high flow and friction environments, and prolong the service life of the coating.	● The construction technology is relatively complicated; ● Coating functionalization; ● Under extreme temperature and humidity conditions, the performance of the coating may be limited, affecting its protective effect.
	Porous carbon spheres [138,139,140,141,142,143]	Artificially synthesized	● High specific surface area; ● Excellent adsorption performance; ● The load of active substances is large; ● Controlled release performance.	● Low yield; ● Poor mechanical properties; ● The preparation process is relatively complicated.		
	Harlow stone [131,132,133,144]	Natural acquisition or artificial synthesis	● Excellent adsorption energy; ● The stability is outstanding; ● Have the ability of ion exchange.	● Low mechanical strength; ● Poor sustained-release effect.		
	Hollow and porous carriers of other ceramics and metal oxides [145,146,147]	Natural acquisition or artificial synthesis	● Excellent compatibility with coatings; ● Excellent mechanical strength.	The synthetic yield is low and the process is complicated.		

#### 3.1.2. Emerging Inorganic Microcapsules

The emerging inorganic microcapsule technology has attracted extensive attention in the field of materials science, especially the new materials such as rodlike cellulose and layered double hydroxide (LDH) [148,149]. These microcapsules have shown remarkable advantages in the application fields of drug delivery, environmental remediation, and catalysis (Table 5). With the development of science and technology and the perfection of characterization technology, researchers have found that many rod-shaped and layered materials can show the characteristics of high loading rate and long-term sustained release of microcapsules.

As a natural polymer material, rod-shaped cellulose has good biocompatibility and biodegradability and can be safely metabolized in the body, reducing the negative impact on the environment. In the drug delivery system, the unique structure of rod-shaped cellulose not only helps to improve the bioavailability of drugs but also can accurately control the drug release rate by adjusting the morphology and surface characteristics of cellulose. This advantage gives it promising application prospects in the treatment of chronic diseases and targeted drug delivery. In addition, the preparation of rod-shaped cellulose is relatively simple, widely available, and low in cost, which is helpful for its popularization in industrial production. For example, Gao [150] and others prepared multi-walled Sn/SnO_2_@carbon hollow nanofibers by electrospinning and carbonization reduction. The unique double-walled pipeline structure made them have a high specific surface area and large internal space. This also increases the active sites of redox reactions and enhances the cycle stability. Multi-walled Sn/SnO_2_@carbon hollow nanofibers can be used in high-performance energy storage systems due to their excellent electrochemical properties. Sheng et al. [151] prepared olive Fe_3_O_4_/carbon hollow microspheres by a simple fibroin-assisted hydrothermal method (Figure 10A). Complex hollow structures were formed by simple and extensible solvothermal method, and Fe_3_O_4_/carbon hollow microspheres had excellent capacity and cycle stability. Cho et al. [152] prepared hollow nanospheres FeSe_2_@GC -rGO hybrid nanofibers, which have hollow morphology (Figure 10B). The synergistic effect of FeSe_2_ and highly conductive rGO matrix makes them have strong electrochemical properties.

Layered double hydroxide (LDH) has always been considered a two-dimensional material with excellent barrier properties. The chemical composition can be expressed as [$M_{1-x}^{2+}M_{x}^{3+}{\left(OH\right)}_{2}]$^X+^[(A^n−^)_x/n_·mH_2_O], where M^2+^ is a divalent metal cation such as Mg^2+^, Ni^2+^, Co^2+^, Zn^2+^, Cu^2+^, etc. M^3+^ is trivalent metal cation such as Al^3+^, Cr^3+^, Fe^3+^ and SC^3+^; An- is an anion, such as inorganic and organic ions such as CO_3_^2−^, NO^3−^, Cl^−^, OH^−^, SO_4_^2−^, PO_4_^3−^, C_6_H_4_(COO)_2_^2−^ and complex ions, so the interlayer spacing of LDHs is different with different inorganic anions. When the x value is between 0.17 and 0.33, that is, the molar ratio is between 0.17 and 0.33, the LDHs with complete structure can be obtained [153,154,155]. In the crystal structure of LDHs, due to the influence of lattice energy minimum effect and lattice orientation effect, metal ions are evenly distributed on the laminate in a certain way, that is, the chemical composition of each tiny structural unit on the laminate remains unchanged [156,157]. Two-dimensional planarity and anion exchange characteristics enable them to effectively block and capture corrosive ions and attract corrosion inhibitors for negative ions. In recent years, with the development of microscopic technology, the characteristics of ion storage between LDH layers have been discovered and widely used as microcapsules in many fields, such as anti-corrosion materials, self-repairing coating materials, and the treatment of anionic dyes in water. In recent years, with the development of microscopic technology, the characteristics of ion storage between LDH layers have been discovered and widely used as microcapsules in many fields, such as anti-corrosion materials, self-repairing coating materials, and the treatment of anionic dyes in water. The research shows that the chemical properties of LDH are stable, and it can effectively load and release drugs through the unique ion exchange and slow-release mechanism, thus improving its targeting and release efficiency. This characteristic makes LDH an ideal choice in water treatment, metal corrosion protection, and other fields. In addition, the synthesis of LDH can be carried out by simple chemical methods, and its structure can be customized to specific functions by means of modification, thus improving its adaptability and efficiency in different applications. For example, Tolstoy [158] and others can quickly hydrolyze the droplets of a mixed solution of NiSO_4_ and FeSO_4_ salts on the surface of an alkaline solution, and obtain open microcapsules with unique vase-like morphology (Figure 10C). The walls of the microcapsules are composed of NiFe_0.3_(OH)_x_ LDH nanocrystals, and the aluminum foil electrode with this microcapsule layer shows active electrocatalytic performance in the oxygen evolution reaction of electrolytic water in an alkaline medium. Xu et al. [159] prepared layered Ni-Fe-Ce-LDH microcapsules by one-step method. By adjusting the doping amount of Ce ion and urea, the aspect ratio of the upper structure of Ni-Fe-Ce-LDH microcapsules was controlled. In addition, the unique 3D Ni-Fe-Ce-LDH microcapsules had excellent OER performance (Figure 10D). Niu et al. [160]. prepared Ni-Fe-LDHs derived from MIL-88A and then exchanged with lanthanide ions to prepare three-dimensional hollow Ni-Fe-Ln-LDHs microcapsules. The microcapsules have a unique porous structure and rich and evenly distributed lanthanide ions unsaturated coordination, which can conveniently, quickly, highly selectively, and sensitively detect the biomarker DPA of Bacillus spores, and can also be applied to the paper vision sensor of integrated smartphones for visual and semi-quantitative DPA.

**Figure 10 ijms-26-01473-f010:**
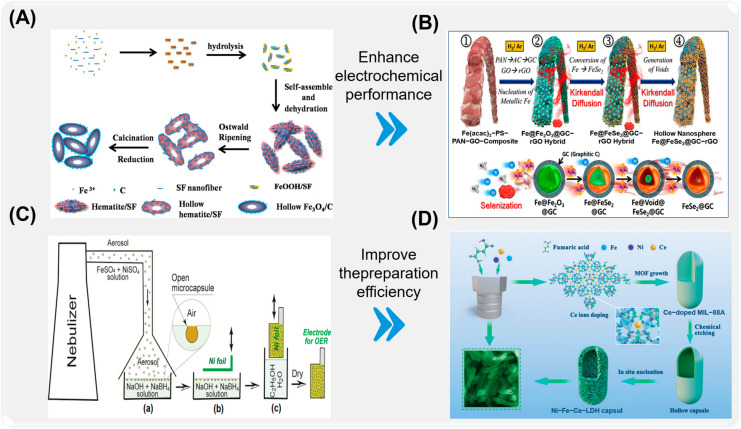
Preparation of different new inorganic microcapsules. (**A**) Preparation of magnetic hollow carbon sphere microcapsules [151]. (**B**) Preparation of graphene oxide hybrid nanofiber microcapsules with excellent electrochemical performance [152]. (**C**) Preparation of new magnetic LDH nanocapsules [158]. (**D**) Preparation of high-efficiency Ni-Fe-Ce-LDH microcapsules [159].

In summary, the emergence of LDH’s new inorganic microcapsule materials broke the researchers’ understanding of hollow microcapsules and expanded the definition of microcapsules. Meanwhile, two-dimensional microcapsule materials show new characteristics and surprising effects in the fields of anti-corrosion coating, environmental remediation, and catalysis. Although the production cost and controllability still need to be further optimized, the emerging inorganic microcapsules still have broad development space in the above fields.

**Table 5 ijms-26-01473-t005:** The following table shows the characteristics of new inorganic microcapsules (hydrotalcite, two-dimensional organometallic framework, hollow carbon fiber, etc.) and their effects on the corrosion resistance of coatings.

Type	Microcapsule Species	Preparation Method	Advantages of Microcapsules	Disadvantages of Microcapsules	Advantages of Composite Coating	Disadvantages of Composite Coating
Novel inorganic microcapsules	Hydrotalcite [158,159,160]	Natural acquisition or artificial synthesis	● Excellent barrier performance; ● Capable of adsorbing corrosive ions; ● Has long-term sustained-release capability.	● Low synthetic yield; ● Insufficient load capacity; ● Poor acid resistance.	● Excellent anti-corrosion performance; ● Excellent thermal stability; ● The specific LDH composite coating has excellent antibacterial properties; ● Excellent barrier performance.	● The synthesis efficiency of LDH is low and the cost is high; ● LDH filler is easy to agglomerate, and the construction process is complicated.
	Two-dimensional metal–organic framework [161,162,163,164]	Artificially synthesized	● High specific surface area; ● Excellent compatibility of resin; ● Surface modification is easy.	● Insufficient tolerance; ● Limited load performance; ● Heavy metals may pollute the environment.	● Lighten coating weight; ● Good compatibility with the coating, increasing the crosslinking density and barrier performance of the coating; ● Give the coating new functions.	● Complexity of synthesis and processing; ● High cost; ● Insufficient mechanical properties; ● May reduce the temperature resistance of the coating.
	Hollow carbon nanofibers [150,151,152,165]	Artificially synthesized	● Higher specific surface area; ● Excellent compatibility of coating; ● Excellent mechanical properties.	● The synthesis process is relatively complicated; ● Poor stability in humid and extreme pH environments; ● High application cost.	● Significantly reduce the coating weight; ● The mechanical properties of the coating are significantly improved; ● Significantly improve thermal stability and corrosion resistance.	● Composite coating has a high cost; ● The preparation process is complicated; ● The conductivity of carbon fiber may accelerate the corrosion of metal substrates.

### 3.2. Inorganic Microcapsule Composite Coating

Inorganic microcapsules have great potential and advantages in the field of anti-corrosion because of their unparalleled thermal stability and chemical stability [166,167]. By embedding inorganic microcapsules in the coating, the long-term protection of metal substrate can be realized. These microcapsules can carry various anti-corrosion compounds and release them slowly under the influence of the external environment, thus forming a lasting anti-corrosion protective layer on the surface of the coating [168]. This slow-release mechanism not only ensures the effective release of preservatives in time but also can dynamically adjust the protective strength according to environmental changes and enhance the overall performance of the coating. In addition, after adding inorganic microcapsules into the coating, the overall physical properties of the coating are improved. The existence of inorganic microcapsules can improve the wear resistance and impact resistance of the coating and enhance its performance in various application environments. The uniform dispersion of microcapsules makes the coating surface smoother and reduces surface defects, thus reducing the invasion rate of corrosive media. The optimization of this structure not only improves the functionality of the coating but also enhances its appearance quality, which meets the dual requirements of esthetics and durability in modern industry. In practical application, the introduction of inorganic microcapsules provides a new idea for the design and development of coating materials. Researchers continue to explore different types of inorganic microcapsules in order to design more targeted coating materials for specific corrosive environments. This customized development process makes the coating not only provide protection under static conditions but also meet various challenges in a dynamic environment. The wide application of inorganic microcapsules will promote the progress of coating technology, open up more application fields, and improve the overall performance and adaptability of anti-corrosion coatings [20].

For example, inorganic nanocapsules include some nanoparticles with porous structures, such as mesoporous SiO_2_, TiO_2_ nanotubes (TNT), halloysite nanotubes (HNT), and calcium carbonate [109,169,170,171,172]. These nanoparticles can not only be used to carry corrosion inhibitors or healing agents to repair the coating but also be used as filling particles to prevent corrosive substances from infiltrating the coating (Figure 11A,B) [173]. For example, Xie et al. [174] proposed a new application of SiO_2_ nanocapsules. In order to protect magnesium alloys, the team doped SiO_2_ nanocapsules containing corrosion inhibitor (NaF) into the pure nickel coating. When magnesium alloy is corroded, magnesium ions react with released fluoride ions to form MgF_2_ precipitation film, which reduces the corrosion rate of magnesium alloy. In addition, the research shows that long tubular nanocapsules have good aerodynamic and hydrodynamic properties under the same load, and have good development potential. Ubaid et al. [175] transferred epoxy monomer and dodecylamine (DDA) into TNT as self-healing agents and corrosion inhibitors, respectively, to prepare inorganic microcapsules. Scanning electron microscope analysis shows that when the coating matrix is damaged, the healing agent can repair the defect. After soaking in salt water for 5 days, the corrosion inhibitor released by microcapsules effectively slowed down the corrosion rate [169]. It is worth noting that HNT, as an important nano-container, is used in intelligent anti-corrosion coatings, and the corrosion inhibitor can be carried and coated only by simple mixing or vacuum impregnation. Xu et al. [176] filled BTA corrosion inhibitor into HNT for active corrosion protection of polybenzoxazine coating. The ultraviolet–visible spectrum analysis shows that HNT has pH-responsive release characteristics. When H^+^ in the environment reaches a suitable concentration, the surface channel of HNT is opened, and the internal corrosion inhibitor is dissolved and diffused to the substrate surface along the concentration gradient, so as to delay the metal corrosion. In the study of inorganic nanocapsules, graphene nanomaterials are often neglected as raw materials for preparing microcapsules. This is because graphene often appears in people’s field of vision in the form of nanosheets, but graphene still has the potential to prepare microcapsules. Li et al. [177] innovatively prepared the structure of the GO microcapsule shell. They use GO as the stabilizer of Pickering emulsion and use the low surface energy generated by GO to drive the self-assembly of microcapsules. In addition, in order to improve the stability of the GO microcapsule shell, polyether amine molecules are used to effectively bond the gap of GO. Javidparvar et al. [178] electrostatically and chemically adsorbed Ce^3+^ ions by GO (which reacts with Ce^3+^ ions to generate hydroxide), thus preparing GO as a suitable nanocapsule. In addition, in order to improve the corrosion resistance of metal surfaces, Wang et al. [179] developed a new multifunctional mesoporous titanium dioxide whisker carrier based on TiO_2_ and SiO_2_, and successfully grafted EDTMPA and imidazoline onto the inner and outer surfaces of mesoporous TiO_2_, thus preparing inorganic microcapsule corrosion inhibition filler (Figure 11C). Due to the synergistic effect of EDTMPA and imidazoline, the composite coating with active filler is 40 times higher than that with pure resin coating (Figure 11D). This novel, simple, and cheap preparation method of surface functional protective coating has a wide practical application prospect.

## 4. Organic–Inorganic Hybrid Microcapsule and Its Application in Anti-Corrosion Coating

### 4.1. Organic and Inorganic Hybrid Microcapsules

Organic–inorganic hybrid microcapsule is a new material system, which combines organic polymer with the inorganic carrier to form microcapsules with unique properties and functions [181,182]. The advantage of this material is that it combines the advantages of organic and inorganic materials and can show good application potential in many fields.

At present, the research of organic–inorganic hybrid microcapsules can focus on the following aspects. First of all, many researchers explore the compatibility and interaction between different organic polymers and inorganic carriers to optimize their properties. By selecting suitable organic modifiers and adjusting their concentration, the properties of microcapsules can be improved and their chemical stability can be maintained. For example, Yang et al. [183] loaded Ag/TiO_2_ nanoparticles onto polyurea microcapsules (PUMC) and used a silane coupling agent to form mixed shell microcapsules (Ag/TiO_2_-PUMC), which solved the aggregation problem of Ag/TiO_2_ nanoparticles when they were mixed with organic matrix. In addition, the organic properties of polyurea microcapsules enhanced the compatibility with the organic matrix (Figure 12A). Zhang et al. [184] grafted cysteamine on the organic layer and cysteamine induced the formation of titanium dioxide, thus forming an inorganic layer on the organic layer and forming organic–inorganic hybrid microcapsules. Cysteamine, as a bridge between the organic layer and the inorganic layer, enhances the stability of the interface and the mechanical properties of microcapsules.

There are a large number of researchers exploring new synthesis methods of organic–inorganic hybrid microcapsules, such as self-assembly and interfacial polymerization. The new synthesis method can effectively reduce production costs, improve production efficiency, and realize large-scale preparation. In addition, developing green synthesis methods to reduce environmental impact and improve sustainability is also an important direction in the future. For example, Li et al. [185] put forward a new method to synthesize organic–inorganic composite double-shell phase-change microcapsules (DLPCMs). By coating paraffin@melamine formaldehyde (MF) microcapsules with atomic layer deposition (ALD), DLPCMs with ZnO as the shell were formed (Figure 12B). The obtained microcapsules solved the shortcomings of poor thermal conductivity and durability of organic shell, and at the same time solved the problems of poor compactness and coverage without shell, which promoted the multifunctional transformation of thermal storage microcapsules. Tian et al. [186] developed a new method to prepare organic–inorganic hybrid microcapsules by covalent crosslinking and biomimetic mineralization into LBL self-assembly, and the prepared microcapsules have higher mechanical stability, thermal stability, and storage stability (Figure 12C); Zhang et al. [187] proposed a new method to prepare organic–inorganic hybrid microcapsules by combining biomimetic mineralization with bioadhesion. The hybrid microcapsules prepared by this method have a hard appearance, stable mechanical properties, and higher mechanical stability and surface reactivity (Figure 12D). In addition, a multienzyme system was constructed by encapsulating three enzymes, which improved the catalytic activity and operational stability. Hou et al. [188] combined poly-dopamine (PDA) and Fe_3_O_4_ nanoparticles onto a CaCO_3_ microparticle template, and prepared organic–inorganic hybrid magnetic microcapsules with negatively charged Fe_3_O_4_ nanoparticles by the hard template-mediated method. This simple method of preparing organic–inorganic hybrid microcapsules provides a certain idea for the simplification and universality of microbial reactors. Moore et al. [189] synthesized organic–inorganic hybrid microcapsules by liquid precursor reaction, which eliminated the use of solid dispersion and had a thicker and stronger shell than other methods. In addition, the team can adjust the preparation of functional microcapsules through different application requirements to achieve tailor-made results.

In addition, most researchers focus on functional research for specific applications. For example, in order to adjust the thermal conductivity of thermal storage materials and expand the application scope of thermal storage microcapsules, Li et al. [190] synthesized the structure of organic–inorganic double-shell microcapsules by in situ polymerization, so as to accurately adjust the thermal conductivity of microcapsules, so that the thermal performance of a single type of microcapsules can meet various use conditions and is suitable for various fields. In order to make the microcapsules have better biocompatibility and can be used for drug delivery and encapsulation of bioactive molecules, Li et al. [191] incubated the prepared sericin microcapsules in supersaturated calcium phosphate solution containing citric acid to prepare organic–inorganic hybrid microcapsules, which have good cell compatibility, good stability, and long storage time, and can be used as a good loading and releasing system for bioactive molecules. In addition, in order to better apply microcapsules to the preparation of functional coatings, Chen [192] and others successfully synthesized a series of double-shell hybrid microcapsules loaded with self-healing agents (linseed oil, LO) by photopolymerization. The synthesized organic–inorganic hybrid microcapsules not only have excellent barrier properties and mechanical properties but also have good thermal stability and can encapsulate LO stably for a long time to avoid failure.

However, organic–inorganic hybrid microcapsules also have some shortcomings and challenges. The introduction of organic matter can improve the properties of microcapsules, but it may lead to a decrease in the chemical stability of carriers, especially under extreme environmental conditions [181,193]. Many organic polymers are easy to degrade under high temperatures or strong acid and alkali conditions, thus affecting the overall performance and service life of microcapsules. Therefore, it is an important research direction to develop organic modifiers with higher durability and stability. However, the complexity of the synthesis process is also a problem that cannot be ignored. Compared with the preparation of a single material, the synthesis of hybrid microcapsules often requires multi-step reactions and precise process control, which leads to an increase in production costs. Moreover, the interaction between different components may also affect the properties of the final material, leading to problems of repeatability and consistency. Therefore, optimizing the synthesis route and simplifying the production process will be an important direction for future research.

### 4.2. Organic–Inorganic Hybrid Microcapsule Composite Coating

Organic–inorganic hybrid microcapsules include microcapsules prepared by compounding inorganic materials such as metal oxides, transition metal oxides, and inorganic salts with resins, polyaniline, graphene, and other materials with excellent performance or special functions [182]. Because hybrid microcapsules fully combine the excellent characteristics of organic materials and inorganic materials, organic–inorganic hybrid microcapsules show excellent mechanical properties, thermal and chemical stability, and permeability resistance [166]. The composite coating prepared by organic–inorganic hybrid microcapsules has better corrosion resistance and will provide long-term protection for equipment working in marine environments.

For example, corrosion inhibitor BTA is added to mesoporous SiO_2_ particles, and a layer of dopamine (PDA) is coated on the surface of silica particles. Then, the prepared inorganic nanocapsules were added to the waterborne alkyd coatings. The release of BTA is controlled by pH-sensitive PDA, and the released BTA and dissolved PDA form a complex with iron oxide to repair defects and prevent further corrosion [194]. Kongparakul et al. [195] prepared organic–inorganic hybrid microcapsules with urea-formaldehyde resin and SiO_2_ as shells, POT, ethanolamine, and diethanolamine as active substances, and combined the microcapsules with epoxy resin to obtain corrosion-resistant epoxy composite coatings with excellent self-repairing ability. In order to further improve the anti-corrosion ability of the coating and endow the coating with wear resistance and self-lubricating properties. Li et al. [196] prepared polyurea (PU) microcapsules with inorganic porous fly ash adsorbing linseed oil (FA-oil) as the core by interfacial polymerization, and the mechanical properties of the microcapsules were significantly improved. A bifunctional epoxy resin coating with self-lubricating and self-repairing properties was prepared by embedding microcapsules into the epoxy resin matrix. Friction and wear tests and electrochemical tests showed that the friction coefficient of the composite coating decreased by 70.2%. At the same time, the impedance modulus of the composite coating is 4 orders of magnitude higher than that of the pure resin coating, showing excellent corrosion resistance.

Most microcapsules can only provide one-time self-repair for the coating, that is, when the coating is damaged, the microcapsules break and release the repairing agent to complete the coating repair, but when the coating is damaged again, the microcapsules no longer have the ability of secondary repair. Therefore, in order to avoid this dilemma, it may be a better solution to prepare a microcapsule that can improve the coating performance and give the metal substrate the ability to actively inhibit corrosion. Some researchers loaded organic–inorganic hybrid microcapsules with active corrosion inhibitors to prepare active anti-corrosion microcapsule fillers, in which the composite coating prepared by microcapsule fillers can continuously release active substances and chemically react on the surface of the substrate, thus forming a dense protective layer and providing long-term protection for the substrate. For example, Wang et al. [197] loaded the slow-release active substance hydroxyethylidene diphosphonic acid (HEDP) inside porous SiO_2_ by vacuum impregnation, and coated the semi-permeable membrane of Lecithin (Lecithin) on the surface of SiO_2_/HEDP intermediate by rotary evaporation technology, thus preparing the organic–inorganic hybrid microcapsule filler capable of continuously releasing the slow-release inhibitor. The electrochemical test shows that the corrosion resistance of the composite coating with active filler is significantly improved, and the impedance modulus of the optimal composite coating is about 3 orders of magnitude higher than that of the pure resin coating.

To further improve the intelligent release of microcapsules, researchers have studied intelligent microcapsules that respond to external conditions such as pH, temperature, ultraviolet rays, and metal ions. For example, He et al. [198] prepared organic–inorganic hybrid microcapsules with pH response and slow release of BTA corrosion inhibitor by using organically modified silicon hydride nanotubes. The inhibitor was transferred into HNT by vacuum impregnation, and then tetraethyl orthosilicate (TEOS) was coated on the surface of HNT to encapsulate the inhibitor. Subsequently, the corrosion resistance of the sample coating was tested by EIS. The test results showed that the corrosion resistance of the composite coating with microcapsules was significantly improved, and the corrosion inhibitor released by nano-containers played an important role in protecting the substrate. The results show that the corrosion will lead to a weak change in local pH in cathode and anode areas. Therefore, the change in pH can also be used as a reliable trigger for metal unprotected systems. In order to further improve the response sensitivity of microcapsules, in another study, Shchukin et al. [199] used SiO_2_, PEI, and benzotriazole to prepare nano-polyelectrolyte (organic–inorganic) hybrid microcapsules with layer-by-layer (LbL) assembly by sol–gel technology, which showed excellent pH response characteristics. Subsequently, the slow-release filler was mixed with resin to prepare the active composite coating for aluminum alloy protection. During the working process, due to the slight change in pH in the corrosion area, the microcapsule channels were opened, and the corrosion inhibitor (benzotriazole) was released to the substrate surface to delay the corrosion of metals.

Above all, the organic–inorganic hybrid microcapsule combines the advantages of two or more materials and has more excellent mechanical properties and chemical stability than the traditional organic microcapsule. In addition, inorganic microcapsules can be modified by using specific organic polymers to give microcapsules response characteristics. Although the above research improves the tolerance and stability of polymer microcapsules through the hybridization of organic–inorganic materials, it also overcomes the functional defects of a single inorganic carrier. However, the shortcomings of poor tolerance of organic polymers in hybrid microcapsules to organic solvents still exist, and the factors of thermal stability, easy aging, and poor binding ability with inorganic materials still limit their application in long-term corrosion-resistant coatings in marine environments.

## 5. Mechanical Stability of Microcapsules

### 5.1. Organic Microcapsules

The mechanical stability of organic microcapsules is the key factor to determine the performance of their protective coatings. This stability is influenced by several factors inherent in polymer wall materials, such as crosslinking density, molecular weight, wall thickness, and flexibility. High crosslinking density improves the mechanical strength and crack resistance of microcapsules, and ensures that microcapsules can withstand mechanical stress in the process of manufacturing and application. Similarly, polymers with higher molecular weight provide higher tensile strength and durability, while maintaining the flexibility required to adapt to substrate deformation. Organic microcapsules can be roughly divided into single-wall structures and double-wall structures. Single-walled microcapsules are usually prepared by simple methods such as spray drying or interfacial polymerization, which is cost-effective and widely used in general applications [200,201]. However, their thin walls make them easy to crack under pressure, leading to premature release of encapsulant. This limits their applicability in harsh applications that require long-term function or resistance to mechanical damage. Double-walled microcapsules solve these limitations by adding additional protective layers. Through advanced technologies such as solvent evaporation or sequential polymerization, this structure improves mechanical stability and provides better packaging efficiency [202]. The outer wall acts as a shock absorber to reduce the possibility of rupture under mechanical or thermal stress. Although their performance has been improved, the manufacture of double-walled systems is more complicated and expensive, which may limit their scalability. Although organic microcapsules are strengthened in many ways, the poor tolerance to organic solvents and temperature sensitivity of organic polymers are still the biggest limitations for their application in extreme environments.

### 5.2. Inorganic Microcapsules

Inorganic microcapsules, including those based on mesoporous silica, kaolin nanotubes, and layered double hydroxide (LDH), have unparalleled mechanical robustness and chemical resistance [133,203,204]. Their rigid frame enables them to withstand extreme environmental conditions, such as high temperatures and corrosive pH value, making them an ideal choice for professional applications. For example, mesoporous silica microcapsules are particularly valuable because of their high specific surface area and adjustable pore size, which is helpful to effectively encapsulate and control the release of active agents [205]. Similarly, halloysite nanotubes have a natural hollow tube structure, excellent mechanical stability, and high packaging efficiency [206]. Microcapsules based on LDH have layered structure and ion exchange ability, and have unique advantages in the application of anionic corrosion inhibitors or functional additives.

Nevertheless, the inherent brittleness and lack of flexibility of inorganic systems are still great limitations. These defects may lead to fracture or damage under mechanical stress and reduce its effectiveness in dynamic environment. In order to solve these problems, researchers explore the design of composite materials by doping flexible polymers or nanoparticles in inorganic shells, thus enhancing their elasticity and adaptability.

### 5.3. Organic–Inorganic Hybrid Microcapsules

Organic–inorganic hybrid microcapsules combine the elasticity of polymer and the rigidity of inorganic materials to form a synergistic system, which overcomes the individual limitations of its components. The organic phase provides flexibility and adaptability to dynamic mechanical stress, while the inorganic phase endows structural integrity, chemical resistance, and thermal stability. For example, titanium dioxide–polyurea mixed microcapsules are famous for their excellent impact resistance and long-term durability [192]. Inorganic titanium dioxide shell provides a strong barrier to prevent mechanical damage and environmental degradation, while polyurea component ensures flexibility and compatibility with various coating substrates [207].

The mixed system shows excellent performance in a protective coating, especially in preventing crack propagation and prolonging the performance life of the barrier. By reducing water inflow and improving corrosion resistance, these microcapsules significantly extend the service life of the coating [208]. In addition, its dual-phase structure can integrate multiple functions, such as self-repair and active corrosion inhibition, making it an ideal choice for high-performance applications in harsh environments.

Although no microcapsule can meet the application requirements of coatings in all environments, organic and organic–inorganic hybrid microcapsules have shown many disadvantages in the application process due to the insufficient characteristics of temperature resistance, ultraviolet resistance, and organic solvent resistance of organic polymers. Moreover, many tests have proved that inorganic microcapsules still maintain excellent stability compared with organic and hybrid microcapsules in harsh environments such as high temperature, high humidity, and high pressure. It is worth noting that the microcapsule carrier with organic as the main body has good plasticity and functional activity, and at the same time, organic and organic–inorganic hybrid microcapsules can also meet the application requirements under various common environmental conditions. Therefore, it is a more sensible strategy to choose the types of microcapsules according to the actual application environment and cost.

## 6. Conclusions and Prospect

Organic polymer protective coating is an important measure to delay metal corrosion. According to The Wall Street Journal reported in May 2023, “In recent ten years, the annual growth rate of anticorrosive coatings in Asia-Pacific region has exceeded 7%, and the growth rate in North America and Europe is not less than 4%”, and the global demand for anticorrosive coatings is still strong [17]. However, the service life of organic coatings is seriously affected by the micropores left by the volatilization of organic solvents, the erosion of a large number of corrosive ions in the ocean, and the parasitism and metabolism of microorganisms in the ocean. Therefore, it is an important strategy to develop coatings with excellent barrier properties, self-repair, and corrosion resistance to prolong the service life of coatings [209]. Many researchers have proved that microcapsules are one of the most effective ways to endow coatings with the above functions. For example, North American coating manufacturers Xuanwei Company, Usarrow Company, and European company Basf Coatings have successively developed microcapsule fillers with different functions [17,90]. Moreover, with the development of technology and mutual learning between disciplines, the function of microcapsules is still improving, the release mode of active substances is becoming more and more intelligent, and the utilization efficiency is getting higher and higher [210].

Although the use of microcapsules has improved the characteristics of materials, enriched the functions of materials, and even achieved changes in the field of materials, not all types of microcapsules have the same development opportunities and potential. Since the invention of microcapsules in the 1940s, great changes have taken place in the materials, preparation methods, and even connotations of microcapsules. For example, hollow or porous spherical carriers that were carried and coated in the early days were defined as microcapsules. Up to now, there are more than 30 kinds of microcapsules classified according to their structures, and nanosheets that can store molecules or even ions have been awarded by researchers as new microcapsules [20,166,211,212]. Meanwhile, the materials of microcapsules have also undergone changes such as paraffin-organic polymers, natural inorganic substances, and hybrid smart microcapsules. The stability of microcapsules is constantly improving, the preparation method is constantly innovating, and the production scale is gradually increasing. Materials with poor stability are gradually eliminated, and complicated processes are optimized or even replaced. In addition, the intelligence and responsiveness of microcapsules have gradually become an important index to measure the performance of products. Therefore, it is a general trend to use inorganic microcapsules with excellent mechanical properties as carriers and choose organic functional materials with stable chemical properties to prepare organic–inorganic hybrid microcapsules in order to realize long life, multifunction, and intelligence of anticorrosive coatings.

## Figures and Tables

**Figure 1 ijms-26-01473-f001:**
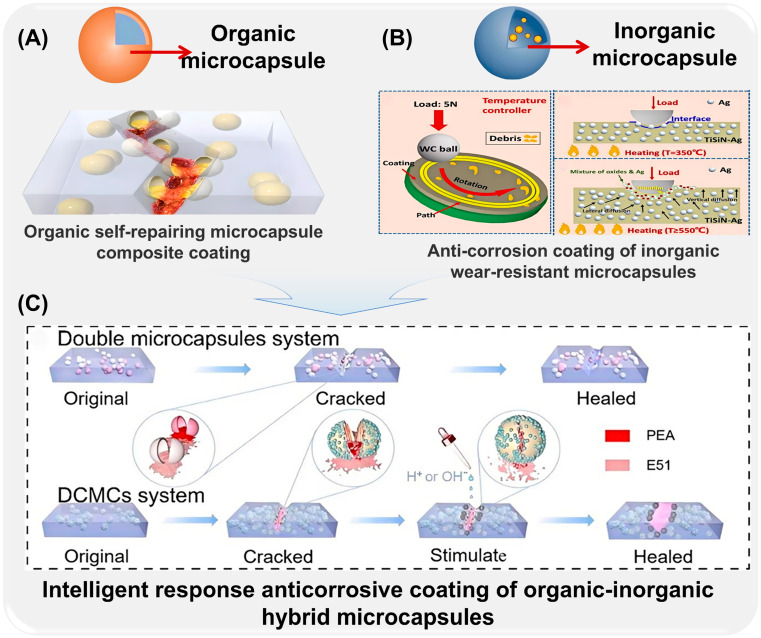
Anti-corrosion mechanism diagram of microcapsule composite coating. (**A**) Mechanism diagram of metal corrosion retarding by organic microcapsule sustained-release inhibitor [28]. (**B**) Corrosion prevention mechanism diagram of inorganic microcapsules [29]. (**C**) Organic and inorganic hybrid microcapsule self-healing anti-corrosion mechanism diagram [30].

**Figure 2 ijms-26-01473-f002:**
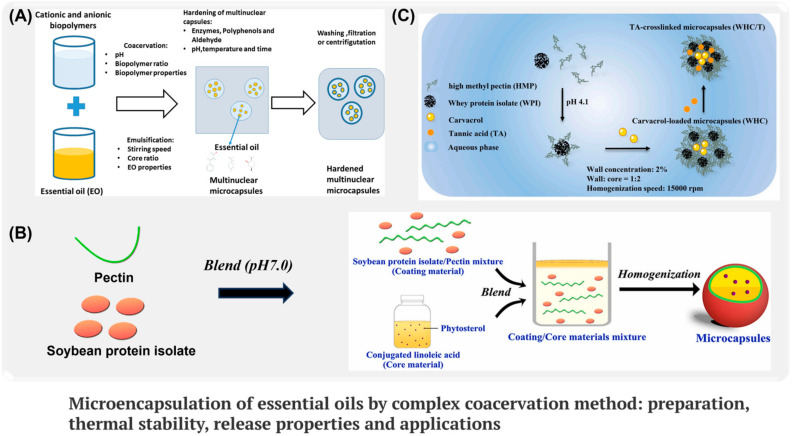
Preparation of single-walled microcapsules with different materials by composite coagulation. (**A**) Preparation of microencapsulated essential oil [57]. (**B**) Preparation of tetrachloroethylene antiseptic microcapsules [61]. (**C**) Preparation of carvacrol sustained-release microcapsules [62].

**Figure 3 ijms-26-01473-f003:**
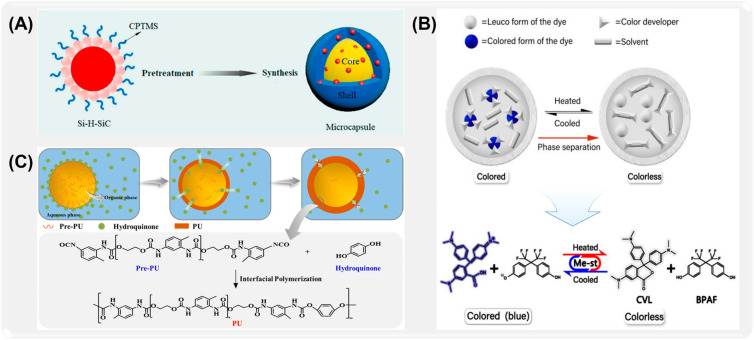
Preparation of different microcapsules by interfacial polymerization. (**A**) Preparation of phase change microcapsules [82]. (**B**) Preparation of new environmental protection (formaldehyde-free) microcapsules [83]. (**C**) Preparation of polyurethane microcapsules [84].

**Figure 4 ijms-26-01473-f004:**
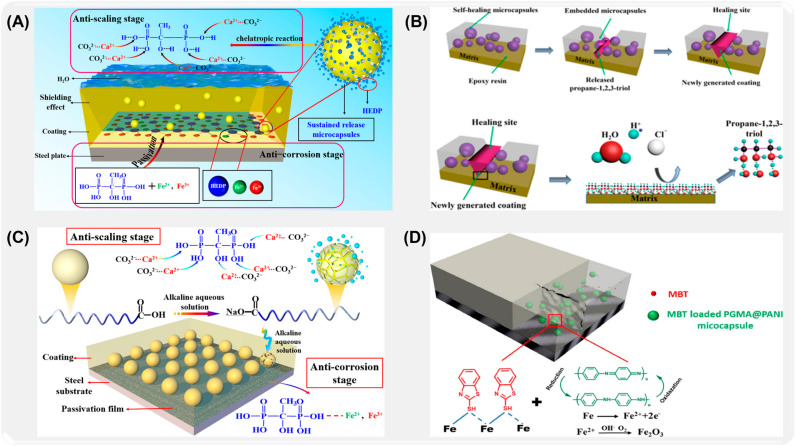
Schematic diagram of single-wall microcapsule composite coating and its anti-corrosion mechanism. (**A**) Anti-corrosion mechanism of long-term sustained-release single-wall microcapsule composite coating [90]. (**B**) Anti-corrosion mechanism of self-repairing single-wall microcapsule composite coating [91]. (**C**) Anti-corrosion mechanism of pH-responsive microcapsule composite coating [92]. (**D**) Double antiseptic mechanism of PGMA@PANI microcapsules loaded with MBT [95].

**Figure 5 ijms-26-01473-f005:**
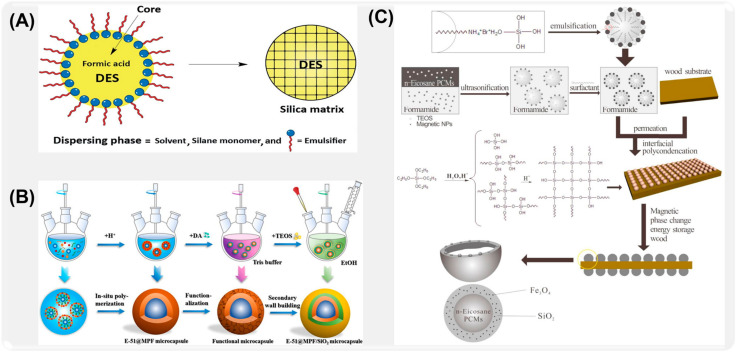
Preparation of double-walled microcapsules by sol–gel: (**A**) Preparation of deep eutectic solvent (DES) microcapsules [101]. (**B**) Preparation of dopamine hydrochloride-SiO_2_ double-wall self-repairing microcapsules [102]. (**C**) Preparation of microcapsules with excellent phase transition stability and temperature control ability [103].

**Figure 6 ijms-26-01473-f006:**
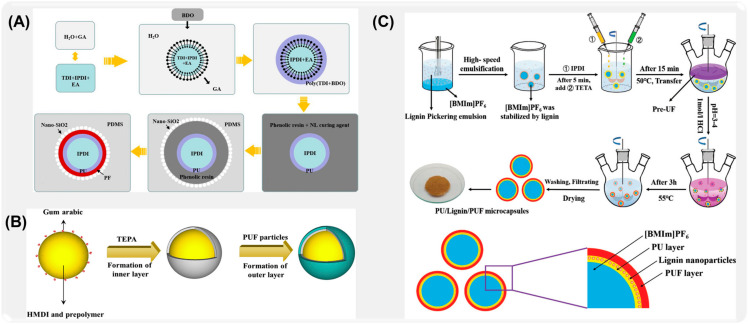
Preparation of double-walled microcapsules by interfacial polymerization and in situ polymerization. (**A**) Preparation of phenolic aldehyde (PF)/polyurethane (PU) double-walled microcapsules [108]. (**B**) Preparation of aqueous double-shell microcapsules [109]. (**C**) Preparation of double-wall self-lubricating microcapsules [110].

**Figure 7 ijms-26-01473-f007:**
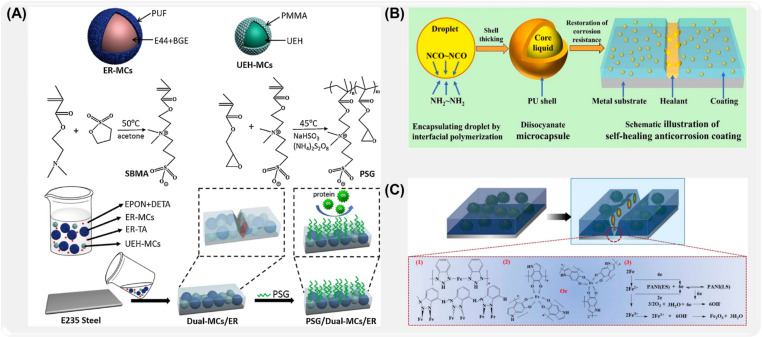
Anti-corrosion mechanism diagram of double-walled microcapsule composite coating. (**A**) Schematic diagram of epoxy resin-curing agent microcapsule multi-component double-walled microcapsule composite coating [117]. (**B**) Anti-corrosion mechanism of HMDI microcapsules prepared by ES-IP technology for anti-corrosion coating [118]. (**C**) Dopamine/polyaniline micro double-wall microcapsule epoxy composite coating [105].

**Figure 8 ijms-26-01473-f008:**
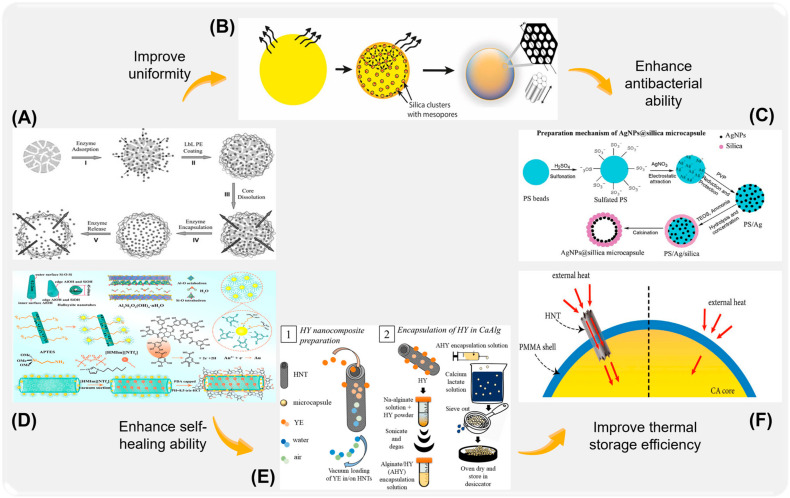
Preparation of inorganic SiO_2_ microcapsules. (**A**) Preparation of mesoporous silica microcapsules [128]. (**B**) Preparation of SiO_2_ by interfacial polymerization [129]. (**C**) Preparation of nano silver-silica (AGPS-SiO_2_) microcapsules [130]. (**D**) Preparation of HNTs/Au high-strength microcapsules [131]. (**E**) Loading YE on HNTs to improve the resistance of microcapsules to cement environment [132]. (**F**) Microcapsules containing kaolin nanotubes (HNTs) were prepared by emulsion copolymerization [133].

**Figure 9 ijms-26-01473-f009:**
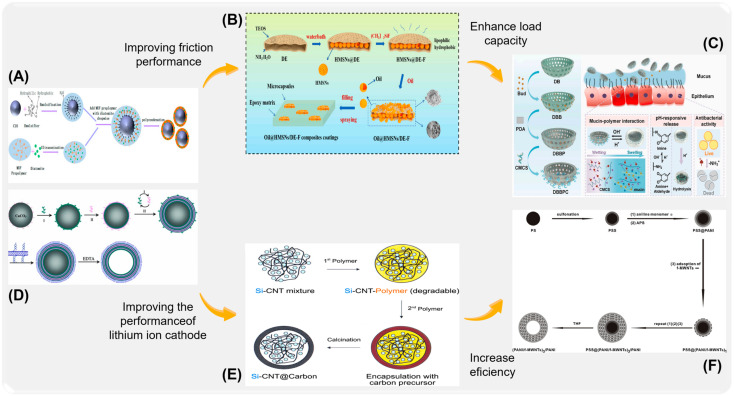
Schematic diagram of preparation and modification of traditional inorganic microcapsules. (**A**) Diatomite inorganic phase-change diatom microcapsules modified by octadecane (C18) and melamine-urea-formaldehyde (MUF) [134]. (**B**) Preparation of fluorosilane-modified hollow mesoporous diatom bio-silicone oil-loaded microcapsules [136]. (**C**) Diatomite microcapsules were prepared by modification of dopamine (PDA) and carboxymethyl chitosan (CMCS) [137]. (**D**) Preparation of carbon nanotube microcapsules by electrostatic self-assembly [138]. (**E**) Preparation of carbon microcapsules of Si-CNT nanocomposites (Si-CNT@C) [139]. (**F**) Simple and efficient preparation method of inorganic microcapsule.

**Figure 11 ijms-26-01473-f011:**
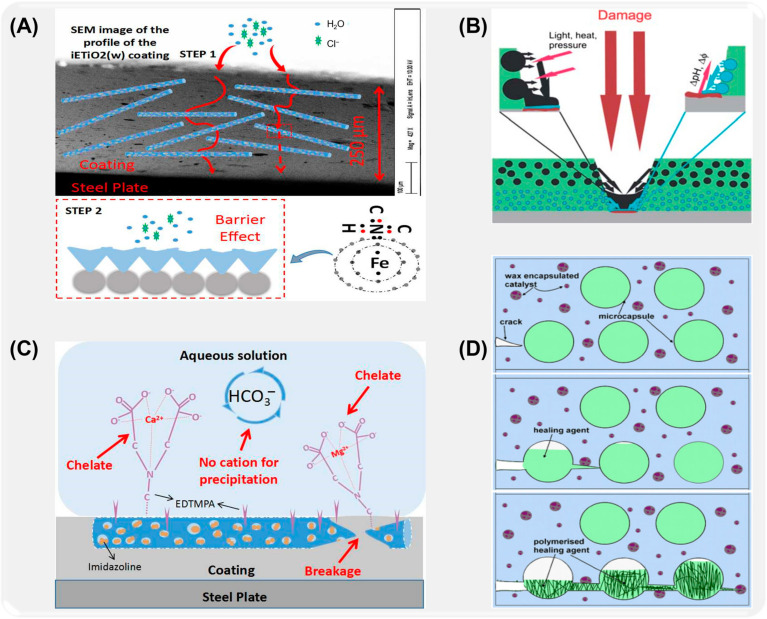
(**A**) Application and mechanism diagram of inorganic microcapsules in the coating. (**A**) Porous slow-release inorganic microcapsule composite coating [180]. (**B**) Inorganic microcapsule self-repairing composite coating [169]. (**C**,**D**) Inorganic anti-corrosion and scale-inhibiting microcapsule composite coating (IETiO_2_) [179].

**Figure 12 ijms-26-01473-f012:**
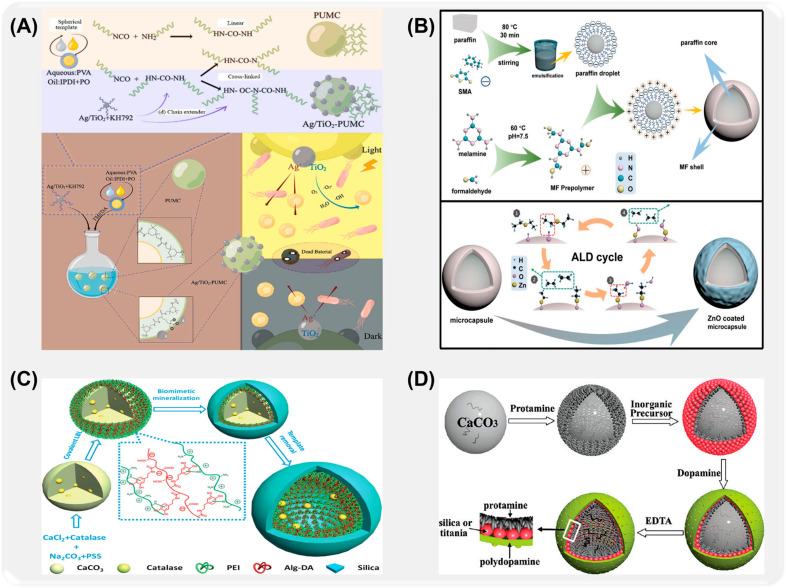
(**A**) Preparation and functional mechanism of different organic–inorganic hybrid microcapsules: preparation method and antibacterial mechanism of Ag/TiO_2_-PUMC [183]. (**B**) Preparation of paraffin @MF hybrid phase-change microcapsules [185]. (**C**) Preparation of active substance-loaded microcapsules (Pei/ALG-DA) Pei/SiO_2_ [186]. (**D**) Design of mineralized bionic hybrid microcapsules (PSI-PDA) [187].

**Table 1 ijms-26-01473-t001:** The classification and characteristics of microcapsules.

Categories of Microcapsules		Advantage	Disadvantage
Organic microcapsule [31,32,33]	Organic single wall [26,34,35,36]	● Mature technology; ● Large-scale preparation and low cost.	● Poor compactness; ● Insufficient tolerance; ● Environmental pollution caused by organic solvents.
	Organic double wall [37,38,39]	● Has classic compactness; ● Excellent stability.	● Poor tolerance; ● Complicated process; ● Environmental pollution caused by organic solvents.
Inorganic microcapsule [40,41,42,43,44]		● Excellent stability and tolerance; ● Excellent mechanical properties.	Insufficient functionality and poor carrying capacity
Organic–inorganic hybrid microcapsule [45,46,47]		● Strong functionality; ● Excellent tolerance.	Complex process and high cost

## Data Availability

Data contained within the article.

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
