# Peer review of "Progress in the Preparation and Applications of Microcapsules for Protective Coatings Against Corrosion"

_ijms, 2025, doi:10.3390/ijms26041473_

Round 1
Reviewer 1 Report
Comments and Suggestions for Authors
This review paper deals with the synthesis methods and coating applications of microcapsules against corrosion. It is recommended to accept this manuscript for publication after some revision on the basis of comments below.
COMMENTS
1.
The title is too broad because this review exclusively deals with coatings against metal surface corrosion. Therefore, the following title is suggested for this paper:
“Progress in the preparation and applications of microcapsules for protective coatings against corrosion”
2.
All the figures are compressed from literature, and as a consequence several details cannot be seen clearly in these figures. It would be better to present enlarged separate figures under each other as A, B, C, D in all the figures.
3.
On page 2, correctly “… titanium dioxide (TiO2) …” and not “titanium dioxide (Ti3C2)”, that is Ti3C2 is absolutely wrong formula for titanium dioxide.
4.
The authors have to clearly emphasize that they description on the preparation methods for microcapsules not exclusively deal with microcapsules synthesized for corrosion protection, but they attempt to review the preparation methods of microcapsules in a more general way, which includes microcapsules prepared for instance for textile, biotechnology, food, pharmaceutical etc. purposes as well.
5.
On page 5, the authors write that “Seeking truth from facts, …”. However, they do not provide any single fact. Because this is a review article, these facts should be included, analyzed and discussed in details in this review.
6.
It is unclear why “WEISS” is written with capital letters on page 6.
7.
on page 8, the title of section 2.21. is incomplete, that is, “2.2.1. Sol-gel method combination”. Combination with what?
8.
For the Tables, full table titles should be provided. Especially, for Table 3, the authors do not provide a professional title.
9.
In the “Conclusion and prospect” section the authors write that “No one can deny that microcapsules are one of the most effective ways to endow coatings with the above functions”. However, not a single example is provided here to support this claim.
10.
In the “Conclusion and prospect” section the authors claim that “… it is a general trend to use inorganic microcapsules …”. However, no any example is provided on any industrial product and application in the whole manuscript. The authors should add existing industrial products and applications of microcapsule based anticorrosive coatings in this review paper.
11.
Last but not least, careful language check, grammar check and compositional mistakes should be checked in the whole manuscript. For instance, there are several incomplete sentences, such as the first sentence below Figure 1 and the second sentence on the bottom paragraph on page 8. On page 9 at the bottom, correctly “For solving these problems, …” (and not “For solve these problems, …”, etc.
Author Response
Reviewer 1
Comment 1. The title is too broad because this review exclusively deals with coatings against metal surface corrosion. Therefore, the following title is suggested for this paper:“Progress in the preparation and applications of microcapsules for protective coatings against corrosion”
Response 1. Thank you for your correction.
The title of the manuscript has been changed from " Preparation and application progress of microcapsules for pro-tective coatings" to " Progress in the preparation and applications of microcapsules for protective coatings against corrosion"
Comment 2. All the figures are compressed from literature, and as a consequence several details cannot be seen clearly in these figures. It would be better to present enlarged separate figures under each other as A, B, C, D in all the figures.
Response 2.Thank you for your correction.
The clarity of the pictures in the manuscript has been improved to ensure that readers can see some details clearly. In addition, some pictures have been rearranged to increase the rationality of illustrations, which can be seen in the manuscript and have been marked.
Comment 3. On page 2, correctly “… titanium dioxide (TiO2)” and not “titanium dioxide (Ti3C2)”, that is Ti3C2 is absolutely wrong formula for titanium dioxide.
Response 3.Thanks for your valuable comments. To ensure the authenticity of the original text, we summarized the article "Go-Ti3C2Two-Dimensional Heterojunction Nano Material for Anti-Corrosion Enhancement of Epoxy Zinc-Rich Coatings.". The modified parts have been marked in blue. Details are as follows: Shen et al. designed a supramolecular heterojunction composite coating. Graphene oxide (GO) nanosheets were grafted with highly conductive MXene (Ti3C2) and added to zinc-rich epoxy resin. The coating has excellent barrier performance and cathodic corrosion inhibition performance.
Comment 4. The authors have to clearly emphasize that they description on the preparation methods for microcapsules not exclusively deal with microcapsules synthesized for corrosion protection, but they attempt to review the preparation methods of microcapsules in a more general way, which includes microcapsules prepared for instance for textile, biotechnology, food, pharmaceutical etc. purposes as well.
Response 4. Thank you for your valuable comments. In this work, we have added the elaboration and analysis of the general advantages of microcapsules in many fields. However, despite our best efforts, due to time constraints, we can't describe all the areas you mentioned in detail. The details are as follows:
Microcapsules with different materials and preparation methods have been explored in many fields, such as anticorrosion, medicine, catalysis, textile, active substance storage and food preservation, and achieved remarkable results. The application of microcapsules in the field of anti-corrosion has also shown new advantages. Commonly used an-ti-corrosion microcapsules are classified according to their material composition and mo-lecular characteristics. Specifically, as shown in Figure 1, they are divided into organic microcapsules, inorganic microcapsules and organic-inorganic hybrid microcapsules. The preparation, surface functionalization, corrosion resistance and response characteristics of various microcapsule composite coatings are reviewed in this work (Figure 1). The preparation methods, functional characteristics and advantages of different microcapsules are fully analyzed, which is of great significance to the preparation of new functional mi-crocapsules and their application in coating field [29,41-43].
- Organic microcapsules and their composite coatings
Encapsulating active substances within organic polymer-based microcapsules is one of the most convenient and efficient methods for their protection and storage [47-49]. Therefore, adding microcapsules to the coating has great potential to enhance the physical properties of the coating and realize the synergy of chemical substances. There are great differences in preparation methods of microcapsules with different functional characteris-tics, materials and morphological structures, among which materials, product microcap-sule structure and production cost are the best and most important factors to determine the preparation methods of microcapsules. Commonly used organic polymers for these mi-crocapsules include polysulfone resin, polyurea (or polyaldehyde) resin, and epoxy resin [17,50,51]. As the characteristics of polymers are different, there are various methods to prepare organic polymer microcapsules, among which the most commonly used methods include interfacial polymerization, phase separation, extrusion and spray drying. With the different requirements for microcapsules in coating anticorrosion, biopharmaceuticals, food processing and other fields and the progress of microcapsule preparation technology, researchers have developed double-walled microcapsules on the basis of traditional pol-ymer single-walled microcapsules [52-54]. Therefore, according to the structural classifica-tion of organic microcapsules, polymer microcapsules can be mainly divided into sin-gle-walled and double-walled microcapsules (Table 2).
Comment 5. On page 5, the authors write that “Seeking truth from facts, …”. However, they do not provide any single fact. Because this is a review article, these facts should be included, analyzed and discussed in details in this review.
Response 5. Thank you for your correction. The content and logic of the manuscript introduction have been improved. The modified parts have been marked in blue. Details are as follows:
2.1.2. Spray drying process
The advantages of spray drying are evident. First of all, the drying conditions are highly adjustable, the process is rapid, and the product quality can be effectively controlled. Second, spray drying is a straightforward and operationally simple method, offering high production efficiency and scalability for mass production. Moreover, the equipment allows for flexible adjustments to meet specific requirements, including control over humidity, particle size, solubility, and the activity of encapsulated substances. However, the spray drying equipment is usually large, which requires a lot of initial investment and will produce a lot of heat loss during the drying process. Generally speaking, spray drying may not be the most energy-saving method, but the authors believe that the preparation of microcapsule carriers on a large scale has unparalleled advantages, especially suitable for processing heat-sensitive materials that need to be dried quickly. With the continuous progress of technology and the expansion of application fields, spray drying is expected to play an increasingly important role in various industries.
Spray-drying is a method to prepare microcapsules by rapidly drying atomized mate-rial fog in high temperature environment [65,66]. Spray drying mainly includes centrifugal, pressure and airflow spray drying methods [67]. This method can directly dry the emulsi-fier and solution into powder or granular products, and can avoid complicated processes such as evaporation and pulverization (Figure 2B). After spray drying after emulsification, the microcapsules have great differences in functions due to the different molecular struc-tures of emulsifiers. Therefore, this method is widely used in biopharmaceutical, chemical, health care, food and other industries because of its simple process and easy mass pro-duction.
For example, Zhang [56] et al. used gelatin/microcrystalline cellulose (GLT/MCC) complex as co-emulsifier, and maltodextrin as wall material, emulsified and spray-dried to prepare Zanthoxylum bungeanum essential oil microcapsules. GLT forms a colloidal structure by hydrogen bonding and electrostatic interaction between abundant polar amino acid residues and water molecules, and MCC provides better physical stability for the emulsion. Tao et al. [68] prepared hydrophobic microcapsules of grape seed oil loaded with pterostilbene (PTE) by spray drying. The methoxy group and stable phenolic struc-ture of pterostilbene (PTE) enable it to enhance its bioavailability, which makes the aroma of grape seed oil retained by a large degree. Rodríguez-Cortina et al. [69] prepared camellia seed oil (SIO) microcapsules by spray drying, and compared them with those obtained by freeze drying technology. It is found that spray-drying microcapsules have higher embed-ding efficiency and are more economical and effective than freeze-drying microcapsules.
The advantages of spray drying are evident. First of all, the drying conditions are highly adjustable, the process is rapid, and the product quality can be effectively con-trolled. Second, spray drying is a straightforward and operationally simple method, offer-ing high production efficiency and scalability for mass production. Moreover, the equip-ment allows for flexible adjustments to meet specific requirements, including control over humidity, particle size, solubility, and the activity of encapsulated substances. However, the spray drying equipment is usually large, which requires a lot of initial investment and will produce a lot of heat loss during the drying process. Generally speaking, spray drying may not be the most energy-saving method, but the authors believe that the preparation of microcapsule carriers on a large scale has unparalleled advantages, especially suitable for processing heat-sensitive materials that need to be dried quickly. With the continuous progress of technology and the expansion of application fields, spray drying is expected to play an increasingly important role in various industries.
Comment 6. It is unclear why “WEISS” is written with capital letters on page 6.
Response 6. Thank you for your correction. The content and logic of the manuscript introduction have been improved. The modified parts have been marked in blue. Details are as follows: Weiβ et al. [ 77] prepared hydroxypropyl methylcellulose phthalate (HPMCP) into polymer microcapsules in 20%(w/w) sodium sulfate solution by simple coagulation method. As a water-soluble cellulose derivative, HPMCP is stable in acidic environment because of its hydroxypropyl and methyl groups in its molecular structure.
Comment 7. on page 8, the title of section 2.2.1. is incomplete, that is, “2.2.1. Sol-gel method combination”. Combination with what?
Response 7. Thank you for your guidance. The topic of 2.2.1 has been supplemented completely. The details are as follows.
2.2.1. Multiple sol-gel method
The sol-gel method is a widely used technique for preparing microcapsules and is regarded as an advanced approach for constructing double-layer wall structures via a multiple sol-gel method process (Figure 3A) [89]. The first step of this method is to select suitable inner wall materials and prepare single-walled microcapsules containing core materials. The second layer of wall material was deposited on the surface of single-walled microcapsules by sol-gel method and double-walled microcapsules were successfully prepared. At the same time, the heat treatment process can further promote the solidification of gel net-work and improve the thermal stability of microcapsules. The double-walled microcapsules prepared by this method have good thermal stability, high encapsulation n efficiency and excellent mechanical properties. In addition, the controllable size, morphology and chemical properties are the advantages of this method in many fields such as self-lubrication, phase change energy storage, temperature-sensitive discoloration and so on [90].
Comment 8. For the Tables, full table titles should be provided. Especially, for Table 3, the authors do not provide a professional title.
Response 8. Thank you for your valuable comments. All the tables have been sorted out and the completed titles have been supplemented. Details are as follows:
Table 2. This table reports the advantages and potential values of different organic single-walled microcapsules and their anticorrosive composite coatings.
Materials |
type Single-walled microcapsule |
preparation method |
Advantages of method |
Disadvantages of method |
Advantages of composite coating |
Disadvantages of composite coating |
Organic microcapsule |
Single-walled microcapsule |
Single coagulation method [58,76,77,114,115] |
● Excellent uniformity and dispersibility. ● high mechanical strength. ● better sustained release effect. |
● The preparation process is relatively complicated. ● The drug loading is slightly lower. ● The encapsulation efficiency of microcapsules is low. |
● Excellent self-repair and corrosion inhibition performance. ● The filler has good dispersibility and high coating density. ● The active substance acts quickly. |
● The active substance has fast slow release speed and short service life. ● The microcapsules in the coating are easy to break and fail, and the stability is poor. ● Low utilization rate of active substances. |
Complex coacervation method [61-64] |
● the loading efficiency of active substances is high, ● the stability is obviously enhanced, ● Strong adaptability ● The process is relatively simple |
● The preparation process is complicated. ● High cost |
||||
Spray drying process [56,68,69] |
● High yield ● high efficiency and simple control. |
● The equipment used in spray drying is bulky. ● High investment ● The heat loss in the drying process is large. |
||||
In-situ polymerization [57,72-74,116] |
High production efficiency and uniform particle size. |
● Temperature control is very difficult; ● During the polymerization process ● Side reactions may occur, |
||||
Interfacial polymerization [59,81-83] |
● The reaction conditions are mild ● The mechanical strength is high |
● It is difficult to control the reaction rate ● The purity of raw materials and equipment is high. |
Table 3. This table reports the characteristics of different organic double-walled microcapsules and their effects on anticorrosion composite coatings.
|
Organic double-walled microcapsules
|
Sol-gel method [28,91,92] |
● It can be prepared at room temperature. ● The product has high purity and good uniformity. ● The product shape and size are easy to control. ● Environment-friendly. |
● Higher cost. ● Organic matter in raw materials may also be harmful to human health. ● Long coagulation time. ● Complex process and poor compactness. |
● The stability is significantly increased. ● The coating life is significantly enhanced. ● The slow release effect is good, and the utilization efficiency of corrosion inhibitors or active substances is significantly increased. |
● The comprehensive cost of coating increases. ● The coating process is complicated. ● The risk of environmental pollution increases with the increase of organic solvent usage. |
In-situ polymerization-surface loading method [86,94,117] |
● Enhanced stability ● Improve drug loading ● Controlled release ● Reduce monomer loss ● Good dispersibility |
● The preparation process is complicated. ● Higher cost. ● Limitation of reaction conditions. ● Poor compatibility with resin. ● pollute the environment. |
||||
Interfacial polymerization-in-situ polymerization combination [87,96,97] |
● A high degree of structural controllability ● Efficient material utilization rate |
● Poor dispersion ● Temperature and pH are difficult to control. ● Low production efficiency |
||||
Composite emulsification method [88,99,100] |
● The preparation process is relatively simple. ● The active substance has a large load, mild environment and strong adaptability. |
● Stability of difference ● Low production efficiency ● High viscosity and difficult preparation |
able 4. The following table shows the characteristics of traditional inorganic microcapsules (mesoporous SiO2, diatomite, halloysite, etc.) and their effects on the anticorrosion performance of coatings.
Type |
Microcapsule species |
preparation method |
Advantages of microcapsules |
Disadvantages of microcapsules |
Advantages of composite coating |
Disadvantages of composite coating |
Traditional inorganic microcapsules
|
diatomaceous earth [130,136,137] |
Natural acquisition |
● High specific surface area. ● Natural reserves are large and easy to obtain. ● Low cost. |
● Functionalization difficulty ● The shape and size are uncontrollable. ● Uneven particle size ● Insufficient completeness and purity |
● Enhance that barrier property of the coat. ● Reduce coating cost. ● Lighten coating weight. ● Give the coating other functions, such as fire prevention, sound insulation and so on. |
● Excessive addition of diatomite may lead to the decrease of coating adhesion. ● Lack of toughness. |
Attapulgite [127,128,132] |
Natural acquisition or artificial synthesis |
● High specific surface area. ● Stability Analysis of Structures. ● Environmental pollution-free. |
● Low active material loading ● The manual preparation process is complicated. |
● Enhance the mechanical properties of the coating. ● Improve the barrier ability of the coating. ● Excellent sustained release performance. ● The adhesion of the coating is significantly enhanced. ● Anti-impact and wear, suitable for high flow and friction environment, and prolong the service life of the coating. |
● The construction technology is relatively complicated. ● Coating functionalization. ● Under extreme temperature and humidity conditions, the performance of the coating may be limited, affecting its protective effect. |
|
Porous carbon spheres [164-166] [131,139,140] |
artificially synthesized |
● High specific surface area. ● Excellent adsorption performance. ● The load of active substances is large. ● Controlled release performance. |
● Low yield ● Poor mechanical properties ● The preparation process is relatively complicated. |
|||
Harlow stone [129,133-135] |
Natural acquisition or artificial synthesis |
● Excellent adsorption energy. ● The stability is outstanding. ● Have the ability of ion exchange. |
● Low mechanical strength. ● Poor sustained release effect. |
|||
Hollow and porous carriers of other ceramics and metal oxides [167-169] |
Natural acquisition or artificial synthesis |
● Excellent compatibility with coatings. ● Excellent mechanical strength. |
The synthetic yield is low and the process is complicated. |
Table 5. The following table shows the characteristics of new inorganic microcapsules (hydrotalcite, two-dimensional organometallic framework, hollow carbon fiber, etc.) and their effects on the corrosion resistance of coatings.
Novel inorganic microcapsules |
Hydrotalcite [146-148] |
Natural acquisition or artificial synthesis |
● Excellent barrier performance. ● Capable of adsorbing corrosive ions. ● Has long-term sustained release capability. |
● Low synthetic yield. ● Insufficient load capacity. ● Poor acid resistance. |
● Excellent anticorrosion performance ● Excellent thermal stability ● The specific LDH composite coating has excellent antibacterial properties. ● Excellent barrier performance |
● The synthesis efficiency of LDH is low and the cost is high. ● LDH filler is easy to agglomerate, and the construction process is complicated. |
Two-dimensional metal-organic framework [170-173] |
Artificially synthesized |
● High specific surface area. ● Excellent compatibility of resin. ● Surface modification is easy. |
● Insufficient tolerance. ● Limited load performance. ● Heavy metals may pollute the environment. |
● Lighten coating weight ● Good compatibility with the coating, increasing the crosslinking density and barrier performance of the coating. ● Give the coating new functions. |
● Complexity of synthesis and processing ● High cost ● Insufficient mechanical properties ● May reduce the temperature resistance of the coating. |
|
Hollow carbon nanofibers [143-145,174] |
Artificially synthesized |
● Higher specific surface area. ● Excellent compatibility of coating. ● Excellent mechanical properties. |
● The synthesis process is relatively complicated. ● Poor stability in humid and extreme pH environment. ● High application cost. |
● Significantly reduce the coating weight ● The mechanical properties of the coating are significantly improved. ● Significantly improve thermal stability and corrosion resistance. |
● Composite coating has high cost. ● The preparation process is complicated. ● The conductivity of carbon fiber may accelerate the corrosion of metal substrate. |
Comment 9. In the “Conclusion and prospect” section the authors write that “No one can deny that microcapsules are one of the most effective ways to endow coatings with the above functions”. However, not a single example is provided here to support this claim.
Response 9. Although the use of microcapsules has improved the characteristics of materials, en-riched the functions of materials and even achieved changes in the field of materials, not all types of microcapsules have the same development opportunities and potential. Since the invention of microcapsules in the 1940s, great changes have taken place in the materi-als, preparation methods and even connotations of microcapsules. For example, hollow or porous spherical carriers that were carried and coated in the early days were defined as microcapsules. Up to now, there are more than 30 kinds of microcapsules classified ac-cording to their structures, and nanosheets that can store molecules or even ions have been awarded by researchers as new microcapsules. Meanwhile, the materials of microcapsules have also undergone changes such as paraffin-organic polymers-natural inorganic sub-stances-hybrid smart microcapsules. The stability of microcapsules is constantly improving, the preparation method is constantly innovating, and the production scale is gradually increasing. Materials with poor stability are gradually eliminated, and complicated processes are optimized or even replaced. In addition, the intelligence and responsiveness of microcapsules have gradually become an important index to measure the performance of products. Therefore, it is a general trend to use inorganic microcapsules with excellent mechanical properties as carriers and choose organic functional materials with stable chemical properties to prepare organic-inorganic hybrid microcapsules in order to realize long life, multifunction and intelligence of anticorrosive coatings.
Comment 10. the “Conclusion and prospect” section the authors claim that “… it is a general trend to use inorganic microcapsules …”. However, no any example is provided on any industrial product and application in the whole manuscript. The authors should add existing industrial products and applications of microcapsule based anticorrosive coatings in this review paper.
Response 10. Thank you for your guidance. The language in the conclusion has been carefully considered and corrected. At the same time, the authors searched a lot of information and provided information about companies with industrial production of microcapsule coatings. Details are as follows:
Organic polymer protective coating is an important measure to delay metal corrosion. According to The Wall Street Journal reported in May, 2023, "In recent ten years, the annu-al growth rate of anticorrosive coatings in Asia-Pacific region has exceeded 7%, and the growth rate in North America and Europe is not less than 4%", and the global demand for anticorrosive coatings is still strong. However, the service life of organic coatings is seri-ously affected by the micropores left by the volatilization of organic solvents, the erosion of a large number of corrosive ions in the ocean and the parasitism and metabolism of mi-croorganisms in the ocean. Therefore, it is an important strategy to develop coatings with excellent barrier properties, self-repair and corrosion resistance to prolong the service life of coatings. Many researchers have proved that microcapsules are one of the most effective ways to endow coatings with the above functions. For example, North American coating manufacturers Xuanwei Company, Usarrow Company and European company Basf Coatings have successively developed microcapsule fillers with different functions. Moreover, with the development of technology and mutual learning between disciplines, the function of microcapsules is still improving, the release mode of active substances is becoming more and more intelligent, and the utilization efficiency is getting higher and higher.
Comment 11. Last but not least, careful language check, grammar check and compositional mistakes should be checked in the whole manuscript. For instance, there are several incomplete sentences, such as the first sentence below Figure 1 and the second sentence on the bottom paragraph on page 8. On page 9 at the bottom, correctly “For solving these problems, …” (and not “For solve these problems, …”, etc.
Response 11. Thank you for your guidance. All the contents of the manuscript have been checked and some of them have been corrected and marked in blue font.
Details are as follows: Functional microcapsules can be categorized based on their material composition and molecular characteristics. Specifically, they are classified into organic microcapsules, inorganic microcapsules, and organic-inorganic hybrid microcapsules, as summarized in Table 1.
Compared to traditional methods, interfacial polymerization involves a relatively complex process, potentially increasing production costs and operational challenges. Additionally, while this method enables the fabrication of microcapsules with smooth surfaces and small particle sizes, its low synthesis efficiency may limit its suitability for large-scale production.
The multiple sol-gel method is a widely used technique for preparing microcapsules and is regarded as an advanced approach for constructing double-layer wall structures via a multiple sol-gel method process (Figure 3A).
For solving these problems, process optimization and material innovation still need to be further studied, so as to reduce the impact on the environment while giving consideration to efficiency and function, so as to promote the development of microcapsule technology in a more mature and diversified direction.

Reviewer 2 Report
Comments and Suggestions for Authors
The manuscript "Preparation and Application Progress of Microcapsules for Protective Coatings" provides a comprehensive overview of recent advances in the application of microcapsule composites for coating purposes. The review discusses various synthesis techniques for different types of microcapsules, including organic, inorganic, and hybrid systems, as well as their applications in self-healing, antimicrobial, and anti-corrosive coatings. The advantages and disadvantages of microcapsule composite systems are also thoroughly examined. This review serves as an important resource for researchers and professionals working on the development of microcapsule-based coating materials.
However, to enhance the manuscript's relevance and depth, several key references should be cited:
- https://onlinelibrary.wiley.com/doi/book/10.1002/3527608478
- https://onlinelibrary.wiley.com/doi/10.1002/3527608478.ch1
- https://pubs.acs.org/doi/10.1021/acsami.4c02462
I recommend the review for publication after addressing the following comments:
- Correct typographical errors in the manuscript, such as "conductive tita-nium dioxide (Ti3C2)."
- Replace the term “Composite Coagulation” with “Complex Coacervation.”
- Ensure Table 2 is properly formatted for clarity and consistency.
- Improve the quality of the figures for better readability and detail.
- Provide more emphasis on the challenges and future perspectives in this field to enrich the discussion.
The manuscript "Preparation and Application Progress of Microcapsules for Protective Coatings" provides a comprehensive overview of recent advances in the application of microcapsule composites for coating purposes. The review discusses various synthesis techniques for different types of microcapsules, including organic, inorganic, and hybrid systems, as well as their applications in self-healing, antimicrobial, and anti-corrosive coatings. The advantages and disadvantages of microcapsule composite systems are also thoroughly examined. This review serves as an important resource for researchers and professionals working on the development of microcapsule-based coating materials.
However, to enhance the manuscript's relevance and depth, several key references should be cited:
- https://onlinelibrary.wiley.com/doi/book/10.1002/3527608478
- https://onlinelibrary.wiley.com/doi/10.1002/3527608478.ch1
- https://pubs.acs.org/doi/10.1021/acsami.4c02462
I recommend the review for publication after addressing the following comments:
- Correct typographical errors in the manuscript, such as "conductive tita-nium dioxide (Ti3C2)."
- Replace the term “Composite Coagulation” with “Complex Coacervation.”
- Ensure Table 2 is properly formatted for clarity and consistency.
- Improve the quality of the figures for better readability and detail.
- Provide more emphasis on the challenges and future perspectives in this field to enrich the discussion.
Author Response
Reviewer 2
Comment 1. However, to enhance the manuscript's relevance and depth, several key references should be cited:
https://onlinelibrary.wiley.com/doi/book/10.1002/3527608478
https://onlinelibrary.wiley.com/doi/10.1002/3527608478.ch1
https://pubs.acs.org/doi/10.1021/acsami.4c02462
Response 1. Thanks for the valuable literature you provided. We have read it carefully and provided effective quotations, as follows:
- Lobel, B.T.; Baiocco, D.; Al-Sharabi, M.; Routh, A.F.; Zhang, Z.; Cayre, O.J. Current challenges in microcapsule designs and microencapsulation processes: A review. ACS applied materials & interfaces 2024, 16, 40326-40355.
- 51. Ghosh, S.K. Functional coatings and microencapsulation: a general perspective. Functional Coatings: by polymer microencapsulation 2006, 1-28.
- 212. Ghosh, S.K.;Functional Coatings by Polymer Microencapsulation, Wiley‐VCH Verlag GmbH & Co. KGaA 2006, 371.
Comment 2. Correct typographical errors in the manuscript, such as "conductive tita-nium dioxide (Ti3C2)."
Response 2. Thank you for your valuable comments. There are some misunderstandings in the original text because of the problems of expression. In order to ensure the authenticity of the content, we summarize the article "Go-Ti3C2Two-Dimensional Heterojunction Nano Material for Anti-Corrosion Enhancement of Epoxy Zinc-Rich Coatings.". The modified parts have been marked in blue. Details are as follows: Shen et al. designed a supramolecular heterojunction composite coating. Graphene oxide (GO) nanosheets were grafted with highly conductive MXene (Ti3C2) and added to zinc-rich epoxy resin. The coating has excellent barrier performance and cathodic corrosion inhibition performance.
Comment 3. Replace the term “Composite Coagulation” with “Complex Coacervation.”
Response 3. Thank you for your correction. The content of the manuscript introduction have been improved. The full text "Composite Coagulation" has been replaced with "Complex Coacervation". The modified parts have been marked in blue.
Figure 2. (A) Preparation of Essential Oil Microcapsules by Complex coacervation [55]. (B) preparing edible oil powder microcapsules [56] by spray drying. (C) preparing long-acting phosphorescent microcapsules [57] by in-situ polymerization. (D) preparing citral essential oil microcapsules [58] by a simple coagulation method. (E) Preparation of poly-dopamine capsules by droplet-mediated interfacial polymerization [59].
Comment 4. Ensure Table 2 is properly formatted for clarity and consistency.
Response 4. Thank you for your valuable comments.
The original Table 2 has been reorganized and decomposed into "Table 2" and "Table 3" and displayed in the manuscript.
Table 2. This table reports the advantages and potential values of different organic single-walled microcapsules and their anticorrosive composite coatings.
Materials |
type Single-walled microcapsule |
preparation method |
Advantages of method |
Disadvantages of method |
Advantages of composite coating |
Disadvantages of composite coating |
Organic microcapsule |
Single-walled microcapsule |
Single coagulation method [58,76,77,114,115] |
● Excellent uniformity and dispersibility. ● high mechanical strength. ● better sustained release effect. |
● The preparation process is relatively complicated. ● The drug loading is slightly lower. ● The encapsulation efficiency of microcapsules is low. |
● Excellent self-repair and corrosion inhibition performance. ● The filler has good dispersibility and high coating density. ● The active substance acts quickly. |
● The active substance has fast slow release speed and short service life. ● The microcapsules in the coating are easy to break and fail, and the stability is poor. ● Low utilization rate of active substances. |
Complex coacervation method [61-64] |
● the loading efficiency of active substances is high, ● the stability is obviously enhanced, ● Strong adaptability ● The process is relatively simple |
● The preparation process is complicated. ● High cost |
||||
Spray drying process [56,68,69] |
● High yield ● high efficiency and simple control. |
● The equipment used in spray drying is bulky. ● High investment ● The heat loss in the drying process is large. |
||||
In-situ polymerization [57,72-74,116] |
High production efficiency and uniform particle size. |
● Temperature control is very difficult; ● During the polymerization process ● Side reactions may occur, |
||||
Interfacial polymerization [59,81-83] |
● The reaction conditions are mild ● The mechanical strength is high |
● It is difficult to control the reaction rate ● The purity of raw materials and equipment is high. |
Table 3. This table reports the characteristics of different organic double-walled microcapsules and their effects on anticorrosion composite coatings.
|
Organic double-walled microcapsules
|
Sol-gel method [28,91,92] |
● It can be prepared at room temperature. ● The product has high purity and good uniformity. ● The product shape and size are easy to control. ● Environment-friendly. |
● Higher cost. ● Organic matter in raw materials may also be harmful to human health. ● Long coagulation time. ● Complex process and poor compactness. |
● The stability is significantly increased. ● The coating life is significantly enhanced. ● The slow release effect is good, and the utilization efficiency of corrosion inhibitors or active substances is significantly increased. |
● The comprehensive cost of coating increases. ● The coating process is complicated. ● The risk of environmental pollution increases with the increase of organic solvent usage. |
In-situ polymerization-surface loading method [86,94,117] |
● Enhanced stability ● Improve drug loading ● Controlled release ● Reduce monomer loss ● Good dispersibility |
● The preparation process is complicated. ● Higher cost. ● Limitation of reaction conditions. ● Poor compatibility with resin. ● pollute the environment. |
||||
Interfacial polymerization-in-situ polymerization combination [87,96,97] |
● A high degree of structural controllability ● Efficient material utilization rate |
● Poor dispersion ● Temperature and pH are difficult to control. ● Low production efficiency |
||||
Composite emulsification method [88,99,100] |
● The preparation process is relatively simple. ● The active substance has a large load, mild environment and strong adaptability. |
● Stability of difference ● Low production efficiency ● High viscosity and difficult preparation |
Comment 5. Improve the quality of the figures for better readability and detail.
Response 5. Thank you for your guidance. The logic of all the pictures in the text has been improved, and the clarity of the pictures has been optimized to ensure the readability of the manuscript. The changes of text and pictures have been marked in blue font in the manuscript.

Reviewer 3 Report
Comments and Suggestions for Authors
This review article describes the preparation and application of the progress of microcapsules for protective coatings.
To improve the review manuscript, the authors should consider the following modifications.
(1) The authors should discuss in detail the mechanical stability of the organic microcapsules, inorganic microcapsules, and organic-inorganic hybrid microcapsules. Moreover, the authors should discuss the single-walled and double-walled microcapsules and the correlation between the wall thickness and protective coatings performance on the structure of organic and polymer microcapsules.
(2) The authors should discuss in detail the cost analysis for the preparation processes of the organic microcapsules, inorganic microcapsules, and organic-inorganic hybrid microcapsules for comparison purposes.
Therefore, the review manuscript requires major revisions before being accepted in the Special Issue: Obtaining and Studying Properties and Application of Nano/Micro Spherical Structures: Capsules, Micelles and Liposomes in the well-circulated International Journal of Molecular Sciences in its current condition.
Comments on the Quality of English LanguageAbstract: The annual economic loss caused by corrosion accounts for about 2%~4% of GDP, which exceeds the sum of losses caused by fires, floods, droughts, typhoons, and other disasters. Coating is one of the most effective methods to delay metal corrosion. With the development of technology and the intersection of disciplines, functional microcapsules have been applied to anticorrosive coatings, but microcapsules are still being updated. To understand the application progress of microcapsules in anticorrosive coatings, their future development trend is analyzed. The preparation methods, physical and chemical properties, functional characteristics, and development trends of organic, inorganic, and organic-inorganic hybrid microcapsules were described respectively from the perspectives of material and molecular characteristics. Simultaneously, the influence of microcapsules of different materials on the properties of organic coatings is proved by examples. In addition, the research status and future development trends of microcapsule composite coating are introduced in detail. Finally, the great advantages of organic-inorganic hybrid microcapsules modified by functional materials based on natural inorganic materials in improving the utilization efficiency of loaded active substances and prolonging the life of coatings are foreseen.
Author Response
Reviewer 3
Comments 1: Provide more emphasis on the challenges and future perspectives in this field to enrich the discussion.
Response 1:Thank you for your valuable comments. According to the investigation literature and the actual research on the characteristics of microcapsules, the development trend of different microcapsules and the future trend of the industry are analyzed, which are shown in the "Conclusion and Prospect":
- 6. Conclusion and prospect
Organic polymer protective coating is an important measure to delay metal corrosion. According to The Wall Street Journal reported in May, 2023, "In recent ten years, the annu-al growth rate of anticorrosive coatings in Asia-Pacific region has exceeded 7%, and the growth rate in North America and Europe is not less than 4%", and the global demand for anticorrosive coatings is still strong. However, the service life of organic coatings is seri-ously affected by the micropores left by the volatilization of organic solvents, the erosion of a large number of corrosive ions in the ocean and the parasitism and metabolism of mi-croorganisms in the ocean. Therefore, it is an important strategy to develop coatings with excellent barrier properties, self-repair and corrosion resistance to prolong the service life of coatings. Many researchers have proved that microcapsules are one of the most effective ways to endow coatings with the above functions. For example, North American coating manufacturers Xuanwei Company, Usarrow Company and European company Basf Coatings have successively developed microcapsule fillers with different functions. Moreover, with the development of technology and mutual learning between disciplines, the function of microcapsules is still improving, the release mode of active substances is becoming more and more intelligent, and the utilization efficiency is getting higher and higher.
Although the use of microcapsules has improved the characteristics of materials, en-riched the functions of materials and even achieved changes in the field of materials, not all types of microcapsules have the same development opportunities and potential. Since the invention of microcapsules in the 1940s, great changes have taken place in the materi-als, preparation methods and even connotations of microcapsules. For example, hollow or porous spherical carriers that were carried and coated in the early days were defined as microcapsules. Up to now, there are more than 30 kinds of microcapsules classified ac-cording to their structures, and nanosheets that can store molecules or even ions have been awarded by researchers as new microcapsules. Meanwhile, the materials of microcapsules have also undergone changes such as paraffin-organic polymers-natural inorganic sub-stances-hybrid smart microcapsules. The stability of microcapsules is constantly improv-ing, the preparation method is constantly innovating, and the production scale is gradu-ally increasing. Materials with poor stability are gradually eliminated, and complicated processes are optimized or even replaced. In addition, the intelligence and responsiveness of microcapsules have gradually become an important index to measure the performance of products. Therefore, it is a general trend to use inorganic microcapsules with excellent mechanical properties as carriers and choose organic functional materials with stable chemical properties to prepare organic-inorganic hybrid microcapsules in order to realize long life, multifunction and intelligence of anticorrosive coatings.
Comments 2:The authors should discuss in detail the mechanical stability of the organic microcapsules, inorganic microcapsules, and organic-inorganic hybrid microcapsules. Moreover, the authors should discuss the single-walled and double-walled microcapsules and the correlation between the wall thickness and protective coatings performance on the structure of organic and polymer microcapsules.
Response 2:Thank you for your valuable comments. The stability and chemical tolerance of different microcapsules have been deeply analyzed in the manuscript, as follows:
- Mechanical Stability of Microcapsules
5.1. Organic Microcapsules
The mechanical stability of organic microcapsules is the key factor to determine the performance of their protective coatings. This stability is influenced by several factors in-herent in polymer wall materials, such as crosslinking density, molecular weight, wall thickness and flexibility. High crosslinking density improves the mechanical strength and crack resistance of microcapsules, and ensures that microcapsules can withstand me-chanical stress in the process of manufacturing and application. Similarly, polymers with higher molecular weight provide higher tensile strength and durability, while maintaining the flexibility required to adapt to substrate deformation. Organic microcapsules can be roughly divided into single-wall structure and double-wall structure. Single-walled mi-crocapsules are usually prepared by simple methods such as spray drying or interfacial polymerization, which is cost-effective and widely used in general applications. However, their thin walls make them easy to crack under pressure, leading to premature release of encapsulant. This limits their applicability in harsh applications that require long-term function or resistance to mechanical damage. Double-walled microcapsules solve these limitations by adding additional protective layers. Through advanced technologies such as solvent evaporation or sequential polymerization, this structure improves mechanical stability and provides better packaging efficiency. The outer wall acts as a shock absorber to reduce the possibility of rupture under mechanical or thermal stress. Although their performance has been improved, the manufacture of double-walled systems is more com-plicated and expensive, which may limit their scalability. Although organic microcapsules are strengthened in many ways, the poor tolerance to organic solvents and temperature sensitivity of organic polymers are still the biggest limitations for their application in ex-treme environments.
5.2. Inorganic Microcapsules
Inorganic microcapsules, including those based on mesoporous silica, kaolin nano-tubes and layered double hydroxide (LDH), have unparalleled mechanical robustness and chemical resistance. Their rigid frame enables them to withstand extreme environmental conditions, such as high temperature and corrosive pH value, making them an ideal choice for professional applications. For example, mesoporous silica microcapsules are particularly valuable because of their high specific surface area and adjustable pore size, which is helpful to effectively encapsulate and control the release of active agents. Simi-larly, kaolin nanotubes have natural hollow tube structure, excellent mechanical stability and high packaging efficiency. Microcapsules based on LDH have layered structure and ion exchange ability, and have unique advantages in the application of anionic corrosion inhibitors or functional additives.
Nevertheless, the inherent brittleness and lack of flexibility of inorganic systems are still great limitations. These defects may lead to fracture or damage under mechanical stress and reduce its effectiveness in dynamic environment. In order to solve these prob-lems, researchers explore the design of composite materials by doping flexible polymers or nanoparticles in inorganic shells, thus enhancing their elasticity and adaptability.
5.3. Organic-Inorganic Hybrid Microcapsules:
Organic-inorganic hybrid microcapsules combine the elasticity of polymer and the ri-gidity of inorganic materials to form a synergistic system, which overcomes the individual limitations of its components. Organic phase provides flexibility and adaptability to dy-namic mechanical stress, while inorganic phase endows structural integrity, chemical re-sistance and thermal stability. For example, titanium dioxide-polyurea mixed microcap-sules are famous for their excellent impact resistance and long-term durability. Inorganic titanium dioxide shell provides a strong barrier to prevent mechanical damage and envi-ronmental degradation, while polyurea component ensures flexibility and compatibility with various coating substrates.
The mixed system shows excellent performance in protective coating, especially in preventing crack propagation and prolonging the performance life of barrier. By reducing water inflow and improving corrosion resistance, these microcapsules significantly extend the service life of the coating. In addition, its dual-phase structure can integrate multiple functions, such as self-repair and active corrosion inhibition, making it an ideal choice for high-performance applications in harsh environments.
Although no microcapsule can meet the application requirements of coatings in all environments, organic and organic-inorganic hybrid microcapsules have shown many disadvantages in the application process due to the insufficient characteristics of temper-ature resistance, ultraviolet resistance and organic solvent resistance of organic polymers. Moreover, many tests have proved that inorganic microcapsules still maintain excellent stability compared with organic and hybrid microcapsules in harsh environments such as high temperature, high humidity and high pressure. It is worth noting that the microcap-sule carrier with organic as the main body has good plasticity and functional activity, and at the same time, organic and organic-inorganic hybrid microcapsules can also meet the application requirements under various common environmental conditions. Therefore, it is a more sensible strategy to choose the types of microcapsules according to the actual application environment and cost.
Comments 3: The authors should discuss in detail the cost analysis for the preparation processes of the organic microcapsules, inorganic microcapsules, and organic-inorganic hybrid microcapsules for comparison purposes.
Response 3:Thank you for your valuable comments. The cost and other characteristics of different microcapsules have been added to the manuscript and marked in blue font. Details of some modified contents are as follows:
Table 1. The classification and characteristics of microcapsules.
Categories of Microcapsules |
Advantage |
Disadvantage |
|
Organic microcapsule [24-26] |
Organic single wall [27-30] |
● Mature technology ● large-scale preparation and low cost |
● Poor compactness ● insufficient tolerance ● environmental pollution caused by organic solvents |
Organic double wall [31-33] |
● Has classic compactness ● excellent stability |
● Poor tolerance ● complicated process ● environmental pollution caused by organic solvents |
|
Inorganic microcapsule [34-38] |
● Excellent stability and tolerance ● Excellent mechanical properties |
Insufficient functionality and poor carrying capacity |
|
Organic-inorganic hybrid microcapsule [39-41] |
● Strong functionality ● excellent tolerance |
Complex process and high cost |
Through decades of development, great progress has been made in the preparation of microcapsules by complex coacervation. With excellent dispersibility, high mechanical strength and remarkable sustained-release effect, it is widely used in the fields of drug delivery, food processing, environmental remediation, preparation of functional materials and so on. However, the complex preparation process and high cost limit its large-scale application in industry.
Table 4. The following table shows the characteristics of traditional inorganic microcapsules (mesoporous SiO2, diatomite, halloysite, etc.) and their effects on the anticorrosion performance of coatings.
Type |
Microcapsule species |
preparation method |
Advantages of microcapsules |
Disadvantages of microcapsules |
Advantages of composite coating |
Disadvantages of composite coating |
Traditional inorganic microcapsules
|
diatomaceous earth [128-130] |
Natural acquisition |
● High specific surface area. ● Natural reserves are large and easy to obtain. ● Low cost. |
● Functionalization difficulty ● The shape and size are uncontrollable. ● Uneven particle size ● Insufficient completeness and purity |
● Enhance that barrier property of the coat. ● Reduce coating cost. ● Lighten coating weight. ● Give the coating other functions, such as fire prevention, sound insulation and so on. |
● Excessive addition of diatomite may lead to the decrease of coating adhesion. ● Lack of toughness. |
Attapulgite [131-133] |
Natural acquisition or artificial synthesis |
● High specific surface area. ● Stability Analysis of Structures. ● Environmental pollution-free. |
● Low active material loading ● The manual preparation process is complicated. |
● Enhance the mechanical properties of the coating. ● Improve the barrier ability of the coating. ● Excellent sustained release performance. ● The adhesion of the coating is significantly enhanced. ● Anti-impact and wear, suitable for high flow and friction environment, and prolong the service life of the coating. |
● The construction technology is relatively complicated. ● Coating functionalization. ● Under extreme temperature and humidity conditions, the performance of the coating may be limited, affecting its protective effect. |
|
Porous carbon spheres [134-136] [137-139] |
artificially synthesized |
● High specific surface area. ● Excellent adsorption performance. ● The load of active substances is large. ● Controlled release performance. |
● Low yield ● Poor mechanical properties ● The preparation process is relatively complicated. |
|||
Harlow stone [140-143] |
Natural acquisition or artificial synthesis |
● Excellent adsorption energy. ● The stability is outstanding. ● Have the ability of ion exchange. |
● Low mechanical strength. ● Poor sustained release effect. |
|||
Hollow and porous carriers of other ceramics and metal oxides [144-146] |
Natural acquisition or artificial synthesis |
● Excellent compatibility with coatings. ● Excellent mechanical strength. |
The synthetic yield is low and the process is complicated. |

Reviewer 4 Report
Comments and Suggestions for Authors
The article presents an overview of the current knowledge on the production and application of microcapsules for protective coatings.
Overall assessment:
In my opinion, the content of the article is limited to describing the methods of producing microcapsules and reporting what has been studied by other authors.
In my opinion, the Review article fulfills a completely different role.
A Review article should not only describe the current state of research in a given topic area, but also include an analysis or interpretation of a set of basic studies on a given topic, as well as a critical assessment of them. A Review article should be a narrative review that explains the existing knowledge on a given topic, not just what this knowledge is. A Review article should seek an answer to a specific question in the existing scientific literature on a given topic. A Review article should be a meta-analysis that compares and combines the results of previously published studies, usually in order to assess the practical aspects of the issue obtained. Finally, a Review article should discuss controversial or controversial issues, as well as address unresolved problems in a given issue.
Unfortunately, I did not notice anything in this article other than reporting what has been studied. This results in a lack of scientific character in the article. The article presents many diagrams, but they are not properly discussed, and in many cases the analysis of Figures or Tables is left to the reader. I would like to mention that the authors should have the publisher's consent to use figures from other publications. But I leave this issue to the Editors.
Other comments:
From the summary and the Introduction it can be clearly read that this is about the use of microcapsules for anti-corrosion coatings (although in the title these are protective coatings - which is a slightly broader term). However, it is difficult to state this unequivocally from the content of the article. Although the authors sometimes indicate anti-corrosion applications, in a significant number of cases the authors draw attention to completely different properties of coatings or other applications. This does not allow for an unambiguous correlation of methods and types of coatings with anti-corrosion protection. At the same time, if the authors do write about anti-corrosion coatings, it is limited to the formulation "anti-corrosion protection". This is a very broad concept. And from this article it is not possible to find out for what conditions, environments or protection of what surfaces the cited coatings are recommended.
In the Introduction the authors justify the need to review the current knowledge of this research topic, justify why this topic is interesting and what are the developments and new advances in the field of microcapsules for protective coatings. The end of the Introduction contains a description of the scope of the article, but I believe that a more detailed discussion of the structure of the article would be beneficial, i.e. an overview of the main topics that will be discussed and the order in which they will be discussed. This will allow the reader to better understand what to expect from the content of the article. My main comment concerns Table 1 and Figure 1. Although the authors refer to them in the text, they do so in a very concise manner. Both Table 1 and Figure 1 contain a lot of data that is not commented on here. Therefore, I have doubts whether Table 1 and Figure 1 should be included in the Introduction. I believe that they should be referred to in detail or omitted.
In the manuscript, the authors do not use line numbering, which is commonly required in MDPI journals. Makes it difficult to correlate notes with their place in the manuscript.
First sentence in chapter 2: The first sentence of a chapter cannot be a continuation of the chapter title. The sentence must indicate the subject.
Sentence on p. 3-4: “Therefore, according to the structural classification of organic microcapsules, polymer microcapsules can be mainly divided into single-walled and double-walled microcapsules (Table 2)”. – The authors refer here to Table 2, which is only on p. 13-14. Meanwhile, at this point they also include Figure 2, to which they refer only in the following subchapters. I believe that the organization of the text in the manuscript is not appropriate. Figure 2: Each Figure should have a general caption explaining what it represents, and then (A), (B), (C)………… should be explained. It should also be verified whether (A), (B), (C)……. should be written in capital or lower case. But this is not the most important thing. In my opinion, it is incomprehensible why the authors combine several methods or mechanisms into one Figure in Figures 2-9. I do not see such justification even if the methods or mechanisms presented in one Figure belong to the same group. This visualization makes the diagrams difficult to read. Additionally, individual methods or mechanisms are described in separate chapters, which makes it difficult to correlate the text with the diagram presented in Figures when reading the text. I believe that such an organization of the text from Figures is inappropriate and I suggest placing the diagram in the chapter in which it is described.
Chapter 2.3. The authors refer to Figure 4 in the following order 4B→4b→4C→4A. I believe that the organization of the text and Figures is inappropriate here.
Figure 5: The authors refer to Figure 5 in one concise sentence. They also refer to Figure 5A. I believe that such complicated diagrams (which are generally presented in all Figures in this article) require a detailed discussion in the text.
Table 2: The authors do not refer to Table 2 in this place, where it is only placed in the Introduction a few pages earlier. This is incomprehensible. Furthermore, Table 2 contains a lot of information that should be discussed. The reference to this Table 2 in the Introduction is extremely concise and basically concerns only the second column. I would also like to draw attention to the formatting of Table 2 itself – the formatting requires checking, because it is difficult to clearly separate and assign information to different types of microcapsules.
The reference to Table 3 is on page 15, while Table 3 is on page 21. In my opinion, the article requires a serious revision of the formatting and organization of the text, Figures and Tables. I will not indicate such comments in the rest of the review, because they concern the entire content of the manuscript.
Abbreviations should be explained when first used in the text. For example, I state that LDH appears on page 15, and only on page 18 is the explanation of “layered double hydroxide (LDH)”. All abbreviations in the article should be revised.
The description of microcapsules in the form of LDH is imprecise (pp. 18-19). This text does not provide any information on what anions the LDH structures indicated by the authors are based on. Anions, along with trivalent and divalent cations, are the basic components of LDH structures.
Author Response
Reviewer 4
Comments 1: In my opinion, the content of the article is limited to describing the methods of producing microcapsules and reporting what has been studied by other authors.
In my opinion, the Review article fulfills a completely different role.
A Review article should not only describe the current state of research in a given topic area, but also include an analysis or interpretation of a set of basic studies on a given topic, as well as a critical assessment of them. A Review article should be a narrative review that explains the existing knowledge on a given topic, not just what this knowledge is. A Review article should seek an answer to a specific question in the existing scientific literature on a given topic. A Review article should be a meta-analysis that compares and combines the results of previously published studies, usually in order to assess the practical aspects of the issue obtained. Finally, a Review article should discuss controversial or controversial issues, as well as address unresolved problems in a given issue.
Unfortunately, I did not notice anything in this article other than reporting what has been studied. This results in a lack of scientific character in the article. The article presents many diagrams, but they are not properly discussed, and in many cases the analysis of Figures or Tables is left to the reader. I would like to mention that the authors should have the publisher's consent to use figures from other publications. But I leave this issue to the Editors.
Response 1: Thank you for your valuable comments.
Please allow the author team to apologize for insufficient work before. Thank you for your guidance on this manuscript, and the author team agrees with your views on summary and critical work. The author's team adjusted most of the contents of the manuscript, increased the discussion on the characteristics of different microcapsules, and analyzed the potential and future development trend of different microcapsules from multiple dimensions such as sexual performance, preparation methods and economy. I hope to provide useful thinking for the small field of microcapsules and composite coatings. The added and modified contents have been marked in the manuscript, and I look forward to receiving more guidance from you.
Comments 2:From the summary and the Introduction it can be clearly read that this is about the use of microcapsules for anti-corrosion coatings (although in the title these are protective coatings - which is a slightly broader term). However, it is difficult to state this unequivocally from the content of the article. Although the authors sometimes indicate anti-corrosion applications, in a significant number of cases the authors draw attention to completely different properties of coatings or other applications. This does not allow for an unambiguous correlation of methods and types of coatings with anti-corrosion protection. At the same time, if the authors do write about anti-corrosion coatings, it is limited to the formulation "anti-corrosion protection". This is a very broad concept. And from this article it is not possible to find out for what conditions, environments or protection of what surfaces the cited coatings are recommended.
Response 2: Thank you for your guidance. The manuscript does have some shortcomings, such as unclear theme and insufficient depth of introduction and analysis of microcapsule anticorrosion. Therefore, the authors have combed the manuscript, perfected the analysis of microcapsules in the coating and marked it in blue font. It is worth noting that although the application of microcapsules in coatings has special requirements for a small amount of resin materials, the influence of microcapsules on coatings still has similar regular characteristics, that is, microcapsule composite coatings with excellent protective performance can protect all materials and equipment for a long time.
Comments 3:In the Introduction the authors justify the need to review the current knowledge of this research topic, justify why this topic is interesting and what are the developments and new advances in the field of microcapsules for protective coatings. The end of the Introduction contains a description of the scope of the article, but I believe that a more detailed discussion of the structure of the article would be beneficial, i.e. an overview of the main topics that will be discussed and the order in which they will be discussed.
Response 3:Thank you for your correction.
The purpose of this work is to describe microcapsules reasonably in the order of development time and progress potential of microcapsules. However, due to the insufficient description, some misunderstandings may occur. The above deficiencies have been supplemented and corrected in the manuscript, as follows:
Microcapsules with different materials and preparation methods have been explored in many fields, such as anticorrosion, medicine, catalysis, textile, active substance storage and food preservation, and achieved remarkable results. The application of microcapsules in the field of anticorrosion also shows new advantages. Commonly used antiseptic microcapsules are classified according to their material composition and molecular characteristics. As shown in Figure 1, according to the different development history of microcapsules, the preparation, surface functionalization, corrosion resistance and response characteristics of organic microcapsules (Figure 1 A), inorganic microcapsules (Figure 1B) and organic-inorganic hybrid microcapsules with special response ability (Figure 1C) and their composite coatings are reviewed. Although with the improvement of technology, the durability and stability of some microcapsules can no longer meet the needs of high-performance materials in the current field, such as organic single-walled microcapsules, they are still of great significance to explain the mechanism and development of microcapsules. In order to better sort out the characteristics of different microcapsules, the basic physical and chemical properties and characteristics of microcapsules are shown in Table 1 according to the different wall materials of microcapsules. It is of great significance to the preparation of new functional microcapsules and their application in the field of coatings [29,41-43].
Comments 4: This will allow the reader to better understand what to expect from the content of the article. My main comment concerns Table 1 and Figure 1. Although the authors refer to them in the text, they do so in a very concise manner. Both Table 1 and Figure 1 contain a lot of data that is not commented on here. Therefore, I have doubts whether Table 1 and Figure 1 should be included in the Introduction. I believe that they should be referred to in detail or omitted.
Response 4:Thank you for your valuable comments. First of all, we have improved the related contents in Figure 1 and Table 1. Secondly, Figure 1 and Table 1 summarize this work and sort out the venation of microcapsules. Since organic microcapsules were invented in 1936, the concept of microcapsules and inorganic carriers with microcapsule characteristics have been defined as inorganic microcapsules. Meanwhile, with the improvement of technology, more carriers with special functions are designed. According to the literature research and the conclusion of the author's team's practical research, the characteristics of different microcapsules, especially in the coating, are shown in Table 1.
Comments 5:In the manuscript, the authors do not use line numbering, which is commonly required in MDPI journals. Makes it difficult to correlate notes with their place in the manuscript.
Response 5:Thank you for your valuable comments. Clear line numbers have been added to the manuscript.
Comments 6: First sentence in chapter 2: The first sentence of a chapter cannot be a continuation of the chapter title. The sentence must indicate the subject.
Response 6:Thank you for your correction. Wrong sentences have been corrected and inadequate sentences have been perfected. The details are as follows:
Encapsulating active substances within organic polymer-based microcapsules is one of the most convenient and efficient methods for their protection and storage [47-49]. Therefore, adding microcapsules to the coating has great potential to enhance the physical properties of the coating and realize the synergy of chemical substances. There are great differences in preparation methods of microcapsules with different functional characteris-tics, materials and morphological structures, among which materials, product microcap-sule structure and production cost are the best and most important factors to determine the preparation methods of microcapsules. Commonly used organic polymers for these mi-crocapsules include polysulfone resin, polyurea (or polyaldehyde) resin, and epoxy resin [17,50,51]. As the characteristics of polymers are different, there are various methods to prepare organic polymer microcapsules, among which the most commonly used methods include interfacial polymerization, phase separation, extrusion and spray drying. With the different requirements for microcapsules in coating anticorrosion, biopharmaceuticals, food processing and other fields and the progress of microcapsule preparation technology, researchers have developed double-walled microcapsules on the basis of traditional pol-ymer single-walled microcapsules [52-54]. Therefore, according to the structural classifica-tion of organic microcapsules, polymer microcapsules can be mainly divided into sin-gle-walled and double-walled microcapsules (Table 2).
Comments 7: Sentence on p. 3-4: “Therefore, according to the structural classification of organic microcapsules, polymer microcapsules can be mainly divided into single-walled and double-walled microcapsules (Table 2)”. – The authors refer here to Table 2, which is only on p. 13-14. Meanwhile, at this point they also include Figure 2, to which they refer only in the following subchapters.
Response 7:Thank you for your guidance. The authors fully agree with you. The charts in the manuscript have been sorted out and improved, including but not limited to Table 2 and Figure 2. The details have been shown in the manuscript and marked in blue font.
- Organic microcapsules and their composite coatings
Encapsulating active substances within organic polymer-based microcapsules is one of the most convenient and efficient methods for their protection and storage [48-51]. Therefore, adding microcapsules to the coating has great potential to enhance the physical properties of the coating and realize the synergy of chemical substances. As show in table 2, there are great differences in preparation methods of microcapsules with different functional characteristics, materials and morphological structures, among which materials, product microcapsule structure and production cost are the best and most important factors to determine the preparation methods of microcapsules. Commonly used organic polymers for these microcapsules include polysulfone resin, polyurea (or polyaldehyde) resin, and epoxy resin [18,52,53]. As the characteristics of polymers are different, there are various methods to prepare organic polymer microcapsules, among which the most commonly used methods include interfacial polymerization, phase separation, extrusion and spray drying. With the different requirements for microcapsules in coating anticorrosion, biopharmaceuticals, food processing and other fields and the progress of microcapsule preparation technology, researchers have developed double-walled microcapsules on the basis of traditional polymer single-walled microcapsules [54-56]. Therefore, according to the structural classification of organic microcapsules, polymer microcapsules can be mainly divided into single-walled and double-walled microcapsules (Table 2).
As shown in Figure 2A, the complex coagulation method is a method of condensing two emulsions with different charges into capsules by electrostatic attraction [78,79]. According to the molecular characteristics and charge of wall materials, accurate adjustment of environmental pH value is the key to successfully prepare microcapsules by composite solidification method [62]. The microcapsules prepared by this method have the characteristics of high embedding efficiency, good biocompatibility and slow controlled release. The preparation method has been widely used in the fields of drug delivery, food preservation, cosmetics and other fields with strong demand for microcapsules [63]. As shown in Table 2 and 3, due to the different molecular structures of polymers, the properties of microcapsules usually vary. For example, in the research of Li et al. [64], tetrachloroethylene microcapsules were prepared with gelatin, sodium dodecyl sulfate (SDS) and sodium carboxymethyl cellulose (NaCMC) as raw materials by complexation and coagulation (Figure 2B). Microcapsules were gradually formed by the interaction between protonated amino group of protein and deprotonated carboxyl group of polysaccharide. NaCMC, as a kind of water-soluble linear polymer produced by partially replacing the 2,3 and 6 hydroxyl groups of cellulose with hydrophobic carboxymethyl groups, makes microcapsules more compact and has better barrier and thermal stability. At the same time, in order to further prepare environmentally-friendly microcapsule shells, Qi et al. [65] used the combination of free amino and carboxyl groups of Whey Protein Isolation (WPI) and high methoxyl pectin (HMP), and finally crosslinked with tannic acid (TA) to form a wall material encapsulating bioactive compounds (Figure 2C). However, compared with the wider interaction provided by WPI, the functionality of NaCMC becomes more limited.
Comments 8: I believe that the organization of the text in the manuscript is not appropriate. Figure 2: Each Figure should have a general caption explaining what it represents, and then (A), (B), (C)………… should be explained. It should also be verified whether (A), (B), (C)……. should be written in capital or lower case. But this is not the most important thing. In my opinion, it is incomprehensible why the authors combine several methods or mechanisms into one Figure in Figures 2-9. I do not see such justification even if the methods or mechanisms presented in one Figure belong to the same group. This visualization makes the diagrams difficult to read. Additionally, individual methods or mechanisms are described in separate chapters, which makes it difficult to correlate the text with the diagram presented in Figures when reading the text. I believe that such an organization of the text from Figures is inappropriate and I suggest placing the diagram in the chapter in which it is described.
Response 8:Thank you for your correction. The logic and annotations of the pictures in the manuscript have been optimized, as follows:
Figure 1. Anti-corrosion mechanism diagram of microcapsule composite coating: (A) Mechanism diagram of metal corrosion retarding by organic microcapsule sustained-release inhibitor [45]. (B) Corrosion prevention mechanism diagram of inorganic microcapsules [46]. (C) Organic and inorganic hybrid microcapsule self-healing anti-corrosion mechanism diagram [47].
Figure 2. Preparation of Single-walled Microcapsules with Different Materials by Composite Coagulation: (A) Preparation of Microencapsulated Essential Oil [57], (B) Preparation of tetrachloroethylene antiseptic microcapsules [61], (C) Preparation of carvacrol sustained-release microcapsules [62].
Figure 3: Preparation of Different Microcapsules by Interfacial Polymerization: (A) Preparation of Phase Change Microcapsules [71]. (B) Preparation of New Environmental Protection (Formaldehyde-free) Microcapsules [72]. (C) Preparation of Polyurethane Microcapsules [74].
Figure 4. Schematic diagram of single-wall microcapsule composite coating and its anticorrosion mechanism: (A) anticorrosion mechanism of long-term sustained-release single-wall microcapsule composite coating [90]; (B) Anti-corrosion mechanism of self-repairing single-wall microcapsule composite coating [91]; (C) Anti-corrosion mechanism of pH-responsive microcapsule composite coating [92]; (D) Double antiseptic mechanism of PGMA@PANI microcapsules loaded with MBT [93].
Figure 5: Preparation of double-walled microcapsules by sol-gel: (A) Preparation of deep eutectic solvent (DES) microcapsules [112]. (B) Preparation of dopamine hydrochloride-SiO2 double-wall self-repairing microcapsules [99]. (C) Preparation of microcapsules with excellent phase transition stability and temperature control ability [100].
Figure 6: Preparation of double-walled microcapsules by interfacial polymerization and in-situ polymerization: (A) Preparation of phenolic aldehyde (PF)/ polyurethane (PU) double-walled microcapsules [104], (B) Preparation of aqueous double-shell microcapsules[106].(C) Preparation of double-wall self-lubricating microcapsules[105].
Figure 7. Anticorrosion mechanism diagram of double-walled microcapsule composite coating: (a) schematic diagram of epoxy resin-curing agent microcapsule multi-component double-walled microcapsule composite coating [117] ; (B) Anti-corrosion mechanism of HMDI microcapsules prepared B) ES-IP technology for anti-corrosion coating [118]; (C) Dopamine/polyaniline micro double-wall microcapsule epoxy composite coating [101].
Figure 8. Preparation of inorganic SiO2 microcapsules: (A) Preparation of mesoporous silica microcapsules [131]. (B) Preparation of SiO2 by interfacial polymerization [132]. (C) Preparation of nano silver-silica (AGPS-SiO2) microcapsules [133]. (D) Preparation of HNTs/Au high strength microcapsules [140]. (e) loading YE on HNTs to improve the resistance of microcapsules to cement environment [142]. (F) ] Microcapsules containing kaolin nanotubes (HNTs) were prepared by emulsion copolymerization [143].
Figure 9. Schematic diagram of preparation and modification of traditional inorganic microcapsules: (a) diatomite inorganic phase-change diatom microcapsules modified by octadecane (C18) and melamine-urea-formaldehyde (MUF)[128], (B) Preparation of fluorosilane modified hollow mesoporous diatom bio-silicone oil loaded microcapsules[130], (C) Diatomite microcapsules were prepared by modification of dopamine (PDA) and carboxymethyl chitosan (CMCS)[147], (D) Preparation of carbon nanotube microcapsules by electrostatic self-assembly [137], and (E) Preparation of carbon microcapsules of Si-CNT nanocomposites (Si-CNT@C)[138].
Figure 10. Preparation of different new inorganic microcapsules: (a) preparation of magnetic hollow carbon sphere microcapsules[158], (B) preparation of graphene oxide hybrid nanofiber microcapsules with excellent electrochemical performance[159], (C) preparation of new magnetic LDH nanocapsules[150] and and (d) preparation of high-efficiency Ni-Fe-Ce-LDH microcapsules[151].
Figure 11. (A) Application and mechanism diagram of inorganic microcapsules in coating: (A) porous slow-release inorganic microcapsule composite coating [169]; (B) inorganic microcapsule self-repairing composite coating [170]; (C)and (D) inorganic anti-corrosion and scale-inhibiting microcapsule composite coating (IETiO2) [171].
Figure 12. (A) Preparation and functional mechanism of different organic-inorganic hybrid microcapsules: preparation method and antibacterial mechanism of Ag/TiO2-PUMC [183]; (B) preparation of paraffin @MF hybrid phase change microcapsules [185]; (C) preparation of active substance-loaded microcapsules (Pei/ALG-DA) Pei/SiO2 [186], (D) design of mineralized bionic hybrid microcapsules (PSI-PDA) [187].
Comments 9: Chapter 2.3. The authors refer to Figure 4 in the following order 4B→4C→4D→4A. I believe that the organization of the text and Figures is inappropriate here.
Response 9:Thank you for your valuable comments. The errors in Figure 4 has been corrected. The images and comments before and after the correction are as follows:
Before correction:
Figure 4. (A) Double anticorrosion mechanism of PGMA@PANI microcapsules loaded with MBT [102]. (B) Anti-corrosion mechanism of SPUA/PEC-HEDP 70-30 composite coating [103]. (C) Self-healing mechanism of propane-doped 123-triol-loaded microcapsules [104]. (D) Anticorrosion mechanism of SP/SA/PSF/HEDP composite coating [105].
After correction:
Figure 4. Schematic diagram of single-wall microcapsule composite coating and its anticorrosion mechanism: (A) anticorrosion mechanism of long-term sustained-release single-wall microcapsule composite coating [90]; (B) Anti-corrosion mechanism of self-repairing single-wall microcapsule composite coating [91]; (C) Anti-corrosion mechanism of pH-responsive microcapsule composite coating [92]; (D) Double antiseptic mechanism of PGMA@PANI microcapsules loaded with MBT [93].
Comments 10: Figure 5: The authors refer to Figure 5 in one concise sentence. They also refer to Figure 5A. I believe that such complicated diagrams (which are generally presented in all Figures in this article) require a detailed discussion in the text.
Response 10:
Figure 5. Preparation of double-walled microcapsules by sol-gel: (A) Preparation of deep eutectic solvent (DES) microcapsules [112]. (B) Preparation of dopamine hydrochloride-SiO2 double-wall self-repairing microcapsules [99]. (C) Preparation of microcapsules with excellent phase transition stability and temperature control ability [100].
For example, Charlie et al. [112] synthesized deep eutectic solvent (DES) microcapsules using a non-aqueous sol–gel method via oil-in-oil emulsions and silane monomer polycondensation (Figure 5A). The microcapsules were prepared through a cross-linking reaction between tetraethoxyorthosilicate and polydimethoxysiloxane, catalyzed by formic acid. The resulting silica shells exhibited excellent chemical stability and physical durability, making them easier to handle compared to the viscous bulk DES. This method demonstrates the advantages of combining oil-in-oil emulsions with the sol–gel approach for the production of stable and reusable DES-based microcapsules. Zhou [99] made full use of the strong self-assembly characteristics of dopamine hydrochloride, and prepared dopamine hydrochloride-SiO2 double-wall self-repairing microcapsules by secondary wall construction method (Figure 5B). Qian et al. [100] prepared double-walled microcapsules with n- disaccharide as the core and Fe3O4 and SiO2 as the inner shell by sol-gel method (Figure 5C). Not only improves the stability and temperature control ability of the phase change material.
Although the preparation of double-walled microcapsules by sol-gel method shows unique advantages. However, in addition to the shortcomings of complex technology of double-walled microcapsules. The disadvantages of raw materials (such as metal alkoxides), such as high cost, environmental pollution, long drying cycle and possible collapse and contraction of the shell during drying, should also be paid attention to. Nevertheless, the sol-gel method is still of great value in preparing double-walled microcapsules with specific functions, especially in improving the thermal stability and mechanical strength of microcapsules.
Comments 11: Table 2: The authors do not refer to Table 2 in this place, where it is only placed in the Introduction a few pages earlier. This is incomprehensible. Furthermore, Table 2 contains a lot of information that should be discussed. The reference to this Table 2 in the Introduction is extremely concise and basically concerns only the second column. I would also like to draw attention to the formatting of Table 2 itself – the formatting requires checking, because it is difficult to clearly separate and assign information to different types of microcapsules.
Response 11:Thank you for your valuable advice. We have sorted out Table 2 and replaced Table 2 with Tables 2 and 3. At the same time, the format of all tables has been optimized to increase the readability of the manuscript. Specific revisions have been marked in the manuscript.
Table 2. This table reports the advantages and potential values of different organic single-walled microcapsules and their anticorrosive composite coatings.
Materials |
type Single-walled microcapsule |
preparation method |
Advantages of method |
Disadvantages of method |
Advantages of composite coating |
Disadvantages of composite coating |
Organic microcapsule |
Single-walled microcapsule |
Single coagulation method [57-61] |
● Excellent uniformity and dispersibility. ● high mechanical strength. ● better sustained release effect. |
● The preparation process is relatively complicated. ● The drug loading is slightly lower. ● The encapsulation efficiency of microcapsules is low. |
● Excellent self-repair and corrosion inhibition performance. ● The filler has good dispersibility and high coating density. ● The active substance acts quickly. |
● The active substance has fast slow release speed and short service life. ● The microcapsules in the coating are easy to break and fail, and the stability is poor. ● Low utilization rate of active substances. |
Complex coacervation method [62-65] |
● the loading efficiency of active substances is high, ● the stability is obviously enhanced, ● Strong adaptability ● The process is relatively simple |
● The preparation process is complicated. ● High cost |
||||
Spray drying process [66-68] |
● High yield ● high efficiency and simple control. |
● The equipment used in spray drying is bulky. ● High investment ● The heat loss in the drying process is large. |
||||
In-situ polymerization [69-73] |
High production efficiency and uniform particle size. |
● Temperature control is very difficult; ● During the polymerization process ● Side reactions may occur, |
||||
Interfacial polymerization [74-77] |
● The reaction conditions are mild ● The mechanical strength is high |
● It is difficult to control the reaction rate ● The purity of raw materials and equipment is high. |
Table 3. This table reports the characteristics of different organic double-walled microcapsules and their effects on anticorrosion composite coatings.
|
Organic double-walled microcapsules
|
Sol-gel method [29,99,100] |
● It can be prepared at room temperature. ● The product has high purity and good uniformity. ● The product shape and size are easy to control. ● Environment-friendly. |
● Higher cost. ● Organic matter in raw materials may also be harmful to human health. ● Long coagulation time. ● Complex process and poor compactness. |
● The stability is significantly increased. ● The coating life is significantly enhanced. ● The slow release effect is good, and the utilization efficiency of corrosion inhibitors or active substances is significantly increased. |
● The comprehensive cost of coating increases. ● The coating process is complicated. ● The risk of environmental pollution increases with the increase of organic solvent usage. |
In-situ polymerization-surface loading method [101-103] |
● Enhanced stability ● Improve drug loading ● Controlled release ● Reduce monomer loss ● Good dispersibility |
● The preparation process is complicated. ● Higher cost. ● Limitation of reaction conditions. ● Poor compatibility with resin. ● pollute the environment. |
||||
Interfacial polymerization-in-situ polymerization combination [104-106] |
● A high degree of structural controllability ● Efficient material utilization rate |
● Poor dispersion ● Temperature and pH are difficult to control. ● Low production efficiency |
||||
Composite emulsification method [107-109] |
● The preparation process is relatively simple. ● The active substance has a large load, mild environment and strong adaptability. |
● Stability of difference ● Low production efficiency ● High viscosity and difficult preparation |
- Organic microcapsules and their composite coatings
Encapsulating active substances within organic polymer-based microcapsules is one of the most convenient and efficient methods for their protection and storage [48-51]. Therefore, adding microcapsules to the coating has great potential to enhance the physical properties of the coating and realize the synergy of chemical substances. As show in table 2 and 3, there are great differences in preparation methods of microcapsules with different functional characteristics, materials and morphological structures, among which materials, product microcapsule structure and production cost are the best and most important factors to determine the preparation methods of microcapsules. Commonly used organic polymers for these microcapsules include polysulfone resin, polyurea (or polyaldehyde) resin, and epoxy resin [18,52,53]. As the characteristics of polymers are different, there are various methods to prepare organic polymer microcapsules, among which the most commonly used methods include interfacial polymerization, phase separation, extrusion and spray drying. With the different requirements for microcapsules in coating anticorrosion, biopharmaceuticals, food processing and other fields and the progress of microcapsule preparation technology, researchers have developed double-walled microcapsules on the basis of traditional polymer single-walled microcapsules [54-56]. Therefore, according to the structural classification of organic microcapsules, polymer microcapsules can be mainly divided into single-walled and double-walled microcapsules (Table 2).
2.1.1. Complex coacervation method
As shown in Figure 2A, the complex coagulation method is a method of condensing two emulsions with different charges into capsules by electrostatic attraction [78,79]. According to the molecular characteristics and charge of wall materials, accurate adjustment of environmental pH value is the key to successfully prepare microcapsules by composite solidification method [62]. The microcapsules prepared by this method have the characteristics of high embedding efficiency, good biocompatibility and slow controlled release. The preparation method has been widely used in the fields of drug delivery, food preservation, cosmetics and other fields with strong demand for microcapsules [63]. As shown in Table 2, due to the different molecular structures of polymers, the properties of microcapsules usually vary. For example, in the research of Li et al. [64], tetrachloroethylene microcapsules were prepared with gelatin, sodium dodecyl sulfate (SDS) and sodium carboxymethyl cellulose (NaCMC) as raw materials by complexation and coagulation (Figure 2B). Microcapsules were gradually formed by the interaction between protonated amino group of protein and deprotonated carboxyl group of polysaccharide. NaCMC, as a kind of water-soluble linear polymer produced by partially replacing the 2,3 and 6 hydroxyl groups of cellulose with hydrophobic carboxymethyl groups, makes microcapsules more compact and has better barrier and thermal stability. At the same time, in order to further prepare environmentally-friendly microcapsule shells, Qi et al. [65] used the combination of free amino and carboxyl groups of Whey Protein Isolation (WPI) and high methoxyl pectin (HMP), and finally crosslinked with tannic acid (TA) to form a wall material encapsulating bioactive compounds (Figure 2C). However, compared with the wider interaction provided by WPI, the functionality of NaCMC becomes more limited.
Comments 12: The reference to Table 3 is on page 15, while Table 3 is on page 21. In my opinion, the article requires a serious revision of the formatting and organization of the text, Figures and Tables. I will not indicate such comments in the rest of the review, because they concern the entire content of the manuscript.
Response 12:Thank you for your valuable comments. The order of Table 3 has been adjusted in the text. The formatting and organization of the text, charts, and tables in this article have been improved.
Table 4. Thank you for your guidance. All tables including Table 3 have been rearranged and analyzed. For the convenience of readers, tables 2 and 3 are replaced by tables 2 and 3 and tables 4 and 5 respectively, as follows:
Type |
Microcapsule species |
preparation method |
Advantages of microcapsules |
Disadvantages of microcapsules |
Advantages of composite coating |
Disadvantages of composite coating |
Traditional inorganic microcapsules
|
Diatomaceous earth [128-130] |
Natural acquisition |
● High specific surface area. ● Natural reserves are large and easy to obtain. ● Low cost. |
● Functionalization difficulty ● The shape and size are uncontrollable. ● Uneven particle size ● Insufficient completeness and purity |
● Enhance that barrier property of the coat. ● Reduce coating cost. ● Lighten coating weight. ● Give the coating other functions, such as fire prevention, sound insulation and so on. |
● Excessive addition of diatomite may lead to the decrease of coating adhesion. ● Lack of toughness. |
Attapulgite [131-133] |
Natural acquisition or artificial synthesis |
● High specific surface area. ● Stability Analysis of Structures. ● Environmental pollution-free. |
● Low active material loading ● The manual preparation process is complicated. |
● Enhance the mechanical properties of the coating. ● Improve the barrier ability of the coating. ● Excellent sustained release performance. ● The adhesion of the coating is significantly enhanced. ● Anti-impact and wear, suitable for high flow and friction environment, and prolong the service life of the coating. |
● The construction technology is relatively complicated. ● Coating functionalization. ● Under extreme temperature and humidity conditions, the performance of the coating may be limited, affecting its protective effect. |
|
Porous carbon spheres [134-136] [137-139] |
artificially synthesized |
● High specific surface area. ● Excellent adsorption performance. ● The load of active substances is large. ● Controlled release performance. |
● Low yield ● Poor mechanical properties ● The preparation process is relatively complicated. |
|||
Harlow stone [140-143] |
Natural acquisition or artificial synthesis |
● Excellent adsorption energy. ● The stability is outstanding. ● Have the ability of ion exchange. |
● Low mechanical strength. ● Poor sustained release effect. |
|||
Hollow and porous carriers of other ceramics and metal oxides [144-146] |
Natural acquisition or artificial synthesis |
● Excellent compatibility with coatings. ● Excellent mechanical strength. |
The synthetic yield is low and the process is complicated. |
Table 5. The following table shows the characteristics of new inorganic microcapsules (hydrotalcite, two-dimensional organometallic framework, hollow carbon fiber, etc.) and their effects on the corrosion resistance of coatings.
Novel inorganic microcapsules |
Hydrotalcite [150-152] |
Natural acquisition or artificial synthesis |
● Excellent barrier performance. ● Capable of adsorbing corrosive ions. ● Has long-term sustained release capability. |
● Low synthetic yield. ● Insufficient load capacity. ● Poor acid resistance. |
● Excellent anticorrosion performance ● Excellent thermal stability ● The specific LDH composite coating has excellent antibacterial properties. ● Excellent barrier performance |
● The synthesis efficiency of LDH is low and the cost is high. ● LDH filler is easy to agglomerate, and the construction process is complicated. |
Two-dimensional metal-organic framework [153-156] |
Artificially synthesized |
● High specific surface area. ● Excellent compatibility of resin. ● Surface modification is easy. |
● Insufficient tolerance. ● Limited load performance. ● Heavy metals may pollute the environment. |
● Lighten coating weight ● Good compatibility with the coating, increasing the crosslinking density and barrier performance of the coating. ● Give the coating new functions. |
● Complexity of synthesis and processing ● High cost ● Insufficient mechanical properties ● May reduce the temperature resistance of the coating. |
|
Hollow carbon nanofibers [157-160] |
Artificially synthesized |
● Higher specific surface area. ● Excellent compatibility of coating. ● Excellent mechanical properties. |
● The synthesis process is relatively complicated. ● Poor stability in humid and extreme pH environment. ● High application cost. |
● Significantly reduce the coating weight ● The mechanical properties of the coating are significantly improved. ● Significantly improve thermal stability and corrosion resistance. |
● Composite coating has high cost. ● The preparation process is complicated. ● The conductivity of carbon fiber may accelerate the corrosion of metal substrate. |
Comments 13:Abbreviations should be explained when first used in the text. For example, I state that LDH appears on page 15, and only on page 18 is the explanation of “layered double hydroxide (LDH)”. All abbreviations in the article should be revised.
The description of microcapsules in the form of LDH is imprecise (pp. 18-19). This text does not provide any information on what anions the LDH structures indicated by the authors are based on. Anions, along with trivalent and divalent cations, are the basic components of LDH structures.
Response 13:Thank you for your valuable comments. The content and logic of the manuscript introduction have been improved. The modified parts have been marked in blue. Details are as follows:
Layered double hydroxide (LDH) has always been considered as a two-dimensional material with excellent barrier properties. LDH, as a kind of inorganic layered material, shows structural similarity with brucite [Mg(OH)2], and divalent cations are partially replaced by trivalent cations [23-25]. Replacing M2+ with M3+ gives an excessive positive charge, which can be neutralized by intercalating anions and water molecules, thus forming a sandwich structure. Two-dimensional planarity and anion exchange characteristics enable them to effectively block and capture corrosive ions and attract corrosion inhibitors for negative ions. In recent years, with the development of microscopic technology, the characteristics of ion storage between LDH layers have been discovered and widely used as microcapsules in many fields, such as anticorrosion materials, self-repairing coating materials and the treatment of anionic dyes in water.

Round 2
Reviewer 2 Report
Comments and Suggestions for Authors
The manuscript has been thoroughly revised by the authors and now meets the required standards. I am pleased to recommend it for publication in the International Journal of Molecular Sciences (IJMS)
Author Response
Comment 1. The manuscript has been thoroughly revised by the authors and now meets the required standards. I am pleased to recommend it for publication in the International Journal of Molecular Sciences (IJMS).
Response 1. We sincerely appreciate the reviewer's rigorous evaluation and constructive suggestions during the peer-review process. We are honored by the recommendation for publication in IJMS and remain committed to contributing to the advancement of molecular science research.
Reviewer 3 Report
Comments and Suggestions for Authors
The manuscript was revised carefully and greatly improved according to the reviewer’s suggestions. The scientific insights are expressed well in this revised submission. The current revision is recommended for publication in the Special Issue: Obtaining and Studying Properties and Application of Nano/Micro Spherical Structures: Capsules, Micelles and Liposomes in the International Journal of Molecular Sciences.
Author Response
Comment 1. The manuscript was revised carefully and greatly improved according to the reviewer’s suggestions. The scientific insights are expressed well in this revised submission. The current revision is recommended for publication in the Special Issue: Obtaining and Studying Properties and Application of Nano/Micro Spherical Structures: Capsules, Micelles and Liposomes in the International Journal of Molecular Sciences.
Response 1. We sincerely appreciate the reviewer's rigorous evaluation and constructive suggestions during the peer-review process. We are honored by the recommendation for publication in IJMS and remain committed to contributing to the advancement of molecular science research.
Reviewer 4 Report
Comments and Suggestions for Authors
The authors sent a corrected article and responded to all my comments.
Unfortunately, I find that there is an extremely large chaos in the submitted responses. This actually forces us to conduct analysis and compare what they write in the response with what was in the first version of the article and with what is in the current version of the article. In addition, the amount of improvement is so large that it is sufficient to send the article for re-review. This means that the article should be rejected with the possibility of re-submission.
But that is not the most important thing. In general, improving the article did not change its quality considering the expectations that are placed on a Review-type article. I have clearly described these expectations in my comments in comment 1. The content of the authors' responses does not fully correlate with the content of my comments. For example, I indicate the following cases:
- The text has been supplemented with information that is presented in Table 1 and Figure 1. This is basically a duplication of information from their captions. If the authors include such complex diagrams, they require commentary, comparison and discussion, including critical discussion. It is similar with Table 1. There is a lot of information there. The role of a Review article is to discuss the available knowledge on a given topic, not to list diagrams. This supplement is not a discussion of the diagrams presented in Figure 1 and Table 1.
- the response to comment 6 is completely incomprehensible. I have drawn attention to a stylistic and grammatical error concerning the correlation of the chapter title with the first task. In response, the authors include some text consisting of 17 lines.
- the text is still incorrectly organized, e.g. Table 2 is placed earlier than the reference to it in the text. Apart from that, I still see no reason to present several methods in one figure. This makes the figures very poorly legible.
- the authors' explanations to the comment regarding LDH are completely incomprehensible. What the authors have added is actually only a general explanation of what LDH is. Only that this explanation is incomplete. But that is not really what I meant. I reported that this existing state of knowledge regarding LDH shows deficiencies, e.g. from this text it was not possible to find out on which anions these LDHs are based. This has not been added yet. I don't understand how you can discuss LDH without giving all its components.
Author Response
Dear reviewer, your questions have been reasonably answered and revised in the manuscript.
Unfortunately, I find that there is an extremely large chaos in the submitted responses. This actually forces us to conduct analysis and compare what they write in the response with what was in the first version of the article and with what is in the current version of the article. In addition, the amount of improvement is so large that it is sufficient to send the article for re-review. This means that the article should be rejected with the possibility of re-submission. But that is not the most important thing. In general, improving the article did not change its quality considering the expectations that are placed on a Review-type article. I have clearly described these expectations in my comments in comment 1. The content of the authors' responses does not fully correlate with the content of my comments. For example, I indicate the following cases:
Comment 1. - The text has been supplemented with information that is presented in Table 1 and Figure 1. This is basically a duplication of information from their captions. If the authors include such complex diagrams, they require commentary, comparison and discussion, including critical discussion. It is similar with Table 1. There is a lot of information there. The role of a Review article is to discuss the available knowledge on a given topic, not to list diagrams. This supplement is not a discussion of the diagrams presented in Figure 1 and Table 1.
Response 1.
All the authors appreciate your guidance. We have made reasonable improvements by referring to other reviews, but it should be noted that we don't think it is appropriate to fully elaborate and deeply analyze the characteristics and advantages of all anti-corrosion functional microcapsules in this limited work. It may be more reasonable for a huge workload to be presented in "monograph" work. Therefore, we try our best to summarize the characteristics of different types of anti-corrosion microcapsules by using tables, which provides a direction for readers who are interested. We still don't think it is wise to explain all the microcapsules involved in the form in the manuscript.
Comment 2.- the response to comment 6 is completely incomprehensible. I have drawn attention to a stylistic and grammatical error concerning the correlation of the chapter title with the first task. In response, the authors include some text consisting of 17 lines.
Response 2.
Thank you for your correction. All grammar problems have been corrected, The details have been shown in the manuscript and marked in blue font.
Comment 3. - the text is still incorrectly organized, e.g. Table 2 is placed earlier than the reference to it in the text. Apart from that, I still see no reason to present several methods in one figure. This makes the figures very poorly legible.
Response 3. Thank you for your guidance. The errors in the front of the table have been corrected, which are shown in the manuscript and marked in blue font. In addition, different products prepared by various related methods are displayed in the same drawing, which is also recognized in many excellent works, such as "DOI: 10.1002/adma.202406825, DOl:10.1002/adma.202312374" and so on. Therefore, in order to facilitate readers to compare related methods and different products, please allow the use of multi-graph combination to explain the work of other researchers.
Comment 4 - the authors' explanations to the comment regarding LDH are completely incomprehensible. What the authors have added is actually only a general explanation of what LDH is. Only that this explanation is incomplete. But that is not really what I meant. I reported that this existing state of knowledge regarding LDH shows deficiencies, e.g. from this text it was not possible to find out on which anions these LDHs are based. This has not been added yet. I don't understand how you can discuss LDH without giving all its components.
Response 4. The details have been shown in the manuscript and marked in blue font.
Layered double hydroxide (LDH) has always been considered as a two-dimensional material with excellent barrier properties. The chemical composition can be expressed as [X+[(An-)x/n•mH2O], where M2+ is divalent metal cation such as Mg2+, Ni2+, Co2+, Zn2+, Cu2+, etc. M3+ is trivalent metal cation such as Al3+, Cr3+, Fe3+ and SC3+; An- is an anion, such as inorganic and organic ions such as CO32-, NO3-, Cl-, OH-, SO42-, PO43-, C6H4 (COO) 22- and complex ions, so the interlayer spacing of LDHs is different with different inorganic anions. When the x value is between 0.17 and 0.33, that is, the molar ratio is between 0.17 and 0.33, the LDHs with complete structure can be obtained [153-155]. In the crystal structure of LDHs, due to the influence of lattice energy minimum effect and lattice orientation effect, metal ions are evenly distributed on the laminate in a certain way, that is, the chemical composition of each tiny structural unit on the laminate remains unchanged [156,157]. Two-dimensional planarity and anion exchange characteristics enable them to effectively block and capture corrosive ions and attract corrosion inhibitors for negative ions. In recent years, with the development of microscopic technology, the characteristics of ion storage between LDH layers have been discovered and widely used as microcapsules in many fields, such as anticorrosion materials, self-repairing coating materials and the treatment of anionic dyes in water.
